# Prion-induced ferroptosis is facilitated by RAC3

Hao Peng[1], Susanne Pfeiffer[1], Borys Varynskyi[1], Marina Qiu[1], Chanikarn Srinark[1], Xiang Jin[2], Xin Zhang[3], Katie Williams[4], Bradley R. Groveman[4], Simote T. Foliaki[4], Brent Race[4], Tina Thomas[4], Chengxuan Chen[5], Constanze Müller[3], Krisztina Kovács[6], Thomas Arzberger[7], Stefan Momma[8], Cathryn L. Haigh[4] & Joel A. Schick[1]✉

Prions are infectious agents that initiate transmissible spongiform encephalopathies, causing devastating neuronal destruction in Creutzfeldt-Jakob and Kuru disease. Rapid cell death depends on presence of the endogenous prion protein PrP$^C$, but its mechanistic contribution to pathogenesis is unclear. Here we investigate the molecular role of PrP$^C$, reactive oxygen species and lipid metabolism in ferroptosis susceptibility, a regulated cell death process characterized by lipid peroxidation. We discover that elevated expression of the cellular prion PrP$^C$ creates a relaxed oxidative milieu that favors accumulation of unsaturated long-chain phospholipids responsible for ferroptotic death. This condition is sustained by the luminal protein glutathione peroxidase 8, which detoxifies reactive species produced by protein misfolding. Consequently, both PrP$^C$ and infectious Creutzfeldt-Jakob disease (CJD) prions trigger ferroptotic markers and sensitization. This lethality is further enhanced by RAC3, a small GTPase. Depletion of RAC3 is observed solely in pathologically afflicted cortices in CJD patients, revealing a synergistic modulation of lipids and reactive species that drives ferroptosis susceptibility. Together, the results show that PrP$^C$ initially suppresses oxidative stress, attenuates cellular defenses, and establishes a systemic vulnerability to the ferroptotic cascade. These results provide insight into the mechanism underlying regulation of ferroptosis in prion diseases and highlight potential therapeutic targets for diseases involving dysregulated cell death processes.

Prion diseases such as Creutzfeldt-Jakob disease (CJD) in humans and scrapie in sheep are characterized by the conversion of normal cellular prion protein PrP$^C$ into a misfolded, beta-sheet-rich isoform (PrP$^{Sc}$). PrP$^{Sc}$ is thought to promote conversion of PrP$^C$ into the misfolded pathological form that then propagates in the brain and aggregates into the major component of scrapie-associated fibrils[1]. Although the precise mechanism by which prions kill neurons has not been established, it is thought to involve the abnormal accumulation of

[1]Genetics and Cellular Engineering Group, Research Unit Signaling and Translation, Helmholtz Zentrum Munich, Neuherberg, Germany. [2]Ministry of Education Key Laboratory for Ecology of Tropical Islands, College of Life Sciences, Hainan Normal University, Haikou, China. [3]Research Unit Analytical Bio-GeoChemistry, Helmholtz Zentrum Munich, Neuherberg, Germany. [4]Laboratory of Neurological Infections and Immunity/Rocky Mountain Veterinary Branch, National Institutes of Health, Hamilton, MT, USA. [5]Department of Biostatistics and Health Data Science, School of Medicine, Indiana University, Indianapolis, IN, USA. [6]Department of Analytical Chemistry, ELTE Eötvös Loránd University, Pázmány, Budapest, Hungary. [7]Center for Neuropathology and Prion Research, Munich, Germany. [8]Institute of Neurology (Edinger Institute), Goethe University, Frankfurt am Main, Germany. ✉e-mail: joel.schick@helmholtz-munich.de

aggregated PrP$^{Sc}$ [2-4]. The misfolded prion protein can disrupt cellular function, damage cellular membranes, and trigger intracellular signaling pathways that ultimately result in neuronal death [5,6]. Crucially, neurotoxicity requires expression of the native endogenous prion protein, demonstrating its key role in the process [6]. Yet, pinpointing the cause of this toxicity has been difficult, resulting in a complex cell death pattern that has so far remained elusive in mechanistic investigations.

Mesenchymalization is a transitional process that leads to increased sensitivity to ferroptosis [7], a form of regulated cell death that involves iron-dependent lipid peroxidation. A central player in ferroptosis, glutathione peroxidase 4 (GPX4), resists this process via glutathione-mediated detoxification of lipid peroxides. Mesenchymalization elevates proteins involved in lipid metabolism and limits antioxidant defense proteins, which can increase susceptibility. These changes can increase vulnerability to ferroptosis-inducing agents, such as erastin and RSL3. Notably, expression of the prion protein is intricately linked to increased mesenchymal markers and metastatic susceptibility in patients [8,9]. Metastatic transformation is associated with increased membrane fluidity, typically achieved via enrichment of membrane polyunsaturated fatty acid (PUFA)-containing phospholipids. The peroxidation of these lipids, which serve as the primary substrate for ferroptotic oxidative stress (reactive oxygen species, ROS), is a key biochemical process directly linked to prion toxicity [10].

In this study, we investigated the consequences of elevated expression of the major prion protein (PrP$^C$) to uncover a molecular basis for cell death in prion-induced diseases. Although cells with elevated PrP$^C$ were lost with extended culture, short-term conditional expression limited cellular oxidative stress via a closely related glutathione peroxidase in the endoplasmic reticulum, GPX8. This decline of ROS led to a new composition of membrane lipids in which ferroptosis-susceptible PUFA-glycerophospholipids comprised the major component. As a result, PrP$^C$-expressing cells are significantly more sensitive to ferroptotic ROS. A bona fide CJD model showed that the pathological prion PrP$^{Sc}$ sensitizes to ferroptosis in organoids, while PRNP-knockout increased resistance. We moreover identified the small-GTPase RAC3 as a key contributor to prion-induced ferroptosis susceptibility due to its impact on membrane ROS, PUFA levels and drive toward mesenchymalization. Cerebral cortex and synaptic junctions from CJD patient brains show a dramatic loss of RAC3 positivity, indicating that the accumulation of PrP$^{Sc}$ and RAC3 together provokes cellular loss. Together, these results uncover a mechanistic basis for ferroptotic cell death susceptibility in prion-related diseases.

## Results

### PrP$^C$ inhibits endoplasmic ROS

Earlier reports indicated the prion protein can act as a surface ferrireductase to enable iron uptake and increase oxidative stress [11-15], which could potentiate ferroptosis. We sought to substantiate these observations in ferroptosis-sensitive HT-1080 fibrosarcoma cells ectopically expressing full-length PrP$^C$ (PrP$^C$ OE). We measured intracellular labile iron using calcein acetoxymethyl but observed no changes relative to vector-containing controls, whereas ferric ammonium citrate or deferoxamine treatment decreased and increased fluorescence signal, respectively. Ferritin heavy and light chain protein levels and FerroOrange were also unchanged in PrP$^C$ OE, suggesting that iron balance is unaffected by elevating PrP$^C$ (Fig. 1A, Supplementary Fig. 1A). An evolutionary and informatics approach also failed to identify a basis for ferrireductase homology in the major prion protein (Supplementary Fig. 1B, C). Finally, we specifically measured iron concentration using a nondestructive approach with Mössbauer spectroscopy. Mössbauer spectra for control and PrP$^C$ OE samples have same Doublet$_1$ with equal Mössbauer parameters (Supplementary File 1), which correspond best to the Fe$^{3+}$ in ferritin (and/or hemosiderin) and have no significant difference in relative transmissions. Thus, iron is

mainly found as Fe$^{3+}$, and its content was not changed in PrP$^C$ OE samples compared to control. The same Fe$^{3+}$ content in both samples confirms that any significant conversion of Fe$^{3+}$ to Fe$^{2+}$ was not observed in PrP$^C$ OE cells.

We observed that cells constitutively expressing PrP$^C$ lose expression over time, despite selection with a dicistronic puromycin resistance gene, while PRNP (the gene encoding PrP$^c$) knockout cells grow faster (Fig. 1B, Supplementary Fig. 1D). The prion protein has been implicated in cellular stress processes [16], thus we suspected PrP$^C$ OE cells were lost due to increased oxidative stress. Surprisingly, testing dihydrodichlorofluorescein diacetate (DCFH-DA) revealed instead a significant decrease in cytosolic reactive species that rose to the same levels as control cells upon short treatment with the ferroptosis inducer 1S,3R-RSL3 (Fig. 1C), consistent with earlier reports showing that the prion protein protects against oxidative stress [17,18]. We speculated that co-regulated proteins could mitigate ROS in response to PrP$^c$ expression. A linear regression analysis of 18,575 genes co-expressed with PRNP unexpectedly revealed GPX8, a paralog of GPX4, to have the highest degree of correlation of all genes (Pearson = 0.621, P (two-tailed) <0.0001; 1393 lines, Cancer Cell Line Encyclopedia; Fig. 1D). As an endoplasmic reticulum resident protein, GPX8 detoxifies H$_2$O$_2$ generated during disulfide isomerase mediated protein (re-)folding [19-22] and is tightly associated with mesenchymalization processes [23]. Analysis of the top 100 genes associated with PRNP expression by Pearson correlation additionally revealed genes involved in epithelial-mesenchymal transition as the highest associated category (Supplementary Fig. 1E).

We observed primary localization to the ER in cells over-expressing PrP$^C$ (Fig. 1E), suggesting the ROS environment in the ER could be affected [24]. The ER redox state is intricately connected to the maintenance of protein-folding equilibrium [25]. To avoid the afore-mentioned long-term culture effects, we generated an HT-1080 cell line conditional for PrP$^C$ expression (PRNP$^{tet}$; Supplementary Fig. 1F) and tested whether PrP$^C$ influenced luminal H$_2$O$_2$ levels using an endoplasmic-specific H$_2$O$_2$ sensor, Hyper3-ER [26]. After 3 days of doxycycline treatment, we observed a striking loss in basal fluorescence signal and in H$_2$O$_2$-supplemented PRNP$^{tet}$ cells, whereas dithiothreitol uniformly led to lower levels in both treated and untreated cells (Fig. 1F). Due to its association with PRNP, we tested next if ectopic PrP$^C$ also triggered GPX8 accumulation and observed a corresponding increase in protein and glutathione peroxidase activity, measured by increased reduced glutathione consumption (Fig. 1G). Increased GPX8 expression was additionally found to be sufficient to inhibit luminal H$_2$O$_2$ (Fig. 1H) and Hyper-ER fluorescence, while knockdown of GPX8 increased luminal H$_2$O$_2$ (Supplementary Fig. 2A). We next explored if GPX8 directly affects PrP$^C$. Using PrP$^C$ cell surface staining, we observed that GPX8 co-transfection strongly depletes surface PrP$^C$ (Supplementary Fig. 2B). Thus, we investigated if total PrP$^C$ protein was decreased and saw that GPX8 potently diminishes cellular PrP$^C$ signal. This effect was reversed with the addition of the proteosome inhibitor MG132, indicating that GPX8 steers PrP$^C$ proteosomal degradation. Taken together, these results indicate that PrP$^C$ acts to depress cellular and specifically ER H$_2$O$_2$, and this redox balance is controlled by GPX8, which facilitates PrP$^C$ degradation.

### PrP$^C$ expression induces a ferroptosis-sensitive phenotype

To investigate loss of PrP$^C$-overexpressing cells over time, we tested cell death inhibitors against apoptosis, necroptosis and ferroptosis using resazurin as a viability indicator. We infected cells with PrP$^C$-containing lentivirus for 3 days and selected for puromycin resistance for 24 h, counted equal numbers of cells, and treated with inhibitors for 3 days. Only Ferrostatin-1 (Fer-1) and alpha-tocopherol (aToc) significantly protected cell viability, suggesting that ferroptosis susceptibility in PrP$^C$ OE cells drives their continual loss (Fig. 2A). Quantitative PCR also showed significantly higher PRNP expression in Fer-1 and

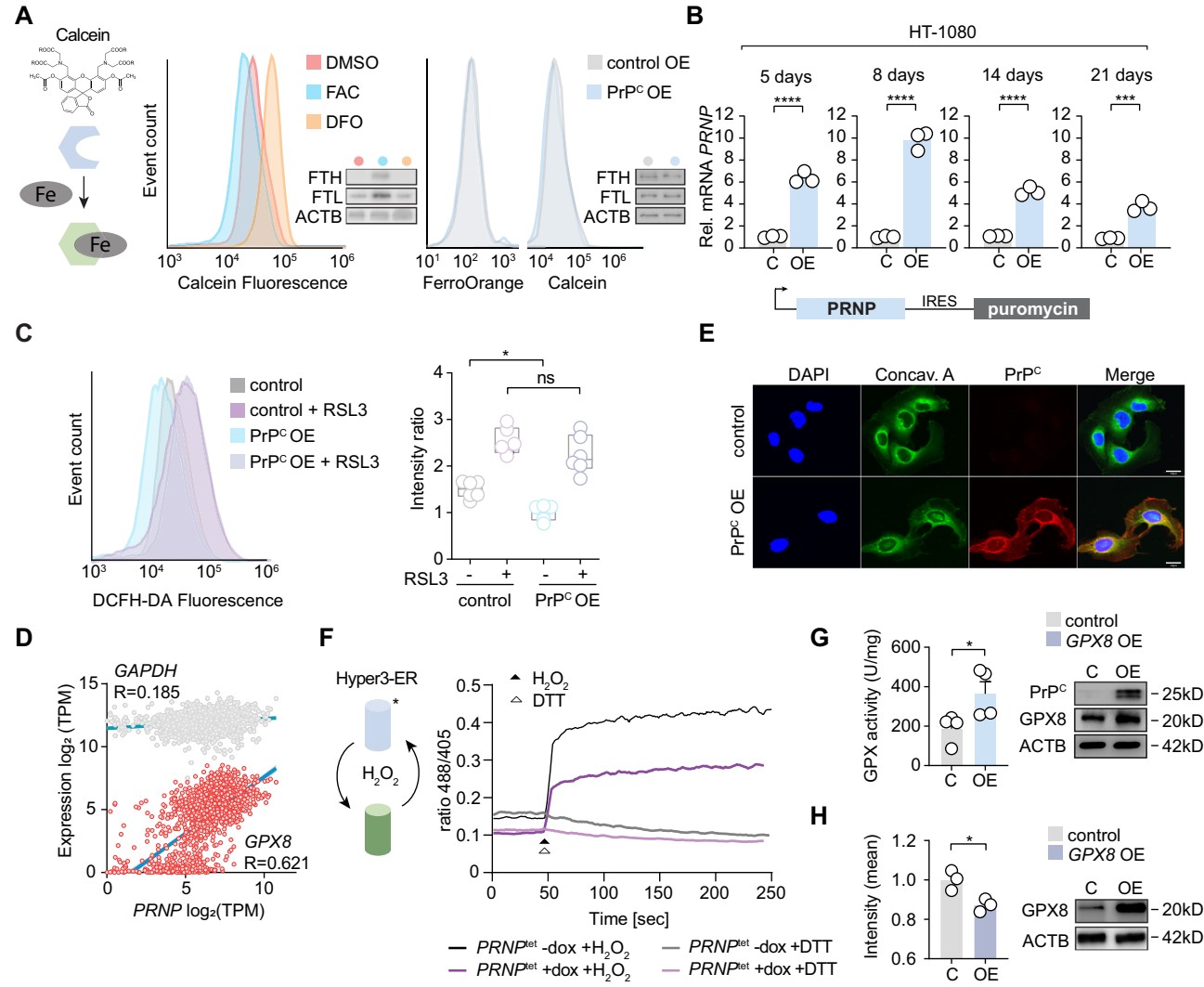

**Fig. 1 | PrP^C represses cellular and endoplasmic ROS. A** Cellular free iron content in empty vector control (control OE) and PrP^C OE HT-1080 cells measured by 5 nM calcein or 1 μM FerroOrange staining. Wild-type HT-1080 treated with 500 μM ferric ammonium citrate (FAC) and 100 μM deferoxamine (DFO) compared to DMSO as the positive and negative control, respectively, a typical FACS histogram is depicted. Insets show ferritin heavy chain (FTH, 21kD), ferritin light chain (FTL, 20kD) and PrP^C levels by Western blot with normalization by beta-actin (ACTB, 42kD). **B** Relative *PRNP* mRNA expression levels change up to 21 days post-infection with a lentiviral expression vector-containing PrP^C (OE) (encoded by *PRNP*) and a separate cistron for puromycin resistance, in 1 μg/mL puromycin-containing media. Total RNA was normalized to empty vector control (C). **C** Cytosolic ROS levels detected by 2,7-Dichlorodihydrofluorescein diacetate (DCFH-DA) using flow cytometry in indicated HT-1080 cells at 7 dpi (days post-infection). A representative flow cytometry histogram of three independent repetitions is depicted (left panel). Boxplot shows (right panel) fluorescence intensity in control and PrP^C OE cells after 0.1 μM RSL3 for 0 h and 2 h, with mean, max and min indicated. **D** Pearson correlation analysis of *GPX8* mRNA expression with *PRNP* mRNA expression ($R = 0.621$; two-tailed $P < 0.0001$; 95% confidence interval 0.587 to 0.651) of 18,575 genes determined for 1393 individual cell lines. *GAPDH* is included as a reference gene. TPM, transcripts per million. **E** Subcellular colocalization of PrP^C together with the endoplasmic reticulum-specific marker concanavalin A (10 μg/mL). Pictures shown are representative results of at least three independent repetitions performed with

similar outcomes (scale bar = 20 μm). **F** Kinetic analysis of ER ROS using Hyper-3 ER fluorescent reporter (emission ratio 488/405) upon PrP^C expression (*PRNP*^tet +dox). Samples were measured by flow cytometry for 1 min to determine baseline intensity before supplementation with DTT (Δ 5 mM) or H_2O_2 (▲ 50 mM). -dox/+dox are samples treated with vehicle or doxycycline, respectively. DTT, dithiothreitol. **G** Glutathione peroxidase (GPX) activity in PrP^C OE compared to empty vector control showing in bar graph. Insets show PrP^C and GPX8 levels by Western in PrP^C OE and control cells. **H** Fluorescence intensity of Hyper-3 ER in HT-1080 cells expressing *GPX8* compared to empty vector control cells. Average fluorescence intensity values (emission ratio 488/405) were determined by flow cytometry. Insets show GPX8 level by Western. **A**, **C** FACS histograms of at least three independent experiments are depicted. Boxplot (**C**) is shown with whiskers min to max. Relative mRNA expression (**B**) is shown as mean ± SEM of $n = 3$ technical replicates. Boxplot data (**C**) is plotted as whiskers min to max, showing all points of $n = 6$ each replicates of duplicate repetitions of the experiment with similar results. Significance was determined by two-tailed *t*-test (**C**). Activity data (**G**) is plotted as representative mean ± SEM of $n = 4$ biological replicates for independent experiments. Intensity (**H**) was calculated versus control, shown as mean ± SEM of $n = 3$ biological replicates. *P*-values of two-tailed *t*-test (**B**, **G**, **H**) or ANOVA multiple comparisons with Tukey post-test are shown for comparisons. *$P < 0.05$, **$P < 0.01$, ***$P < 0.001$, ****$P < 0.0001$. Source data are provided as a Source Data file.

aToc-treated cultures, while a live/dead Calcein-AM/PI stain confirmed cell loss due to lack of viability (Supplementary Fig. 2C, D). Thus, we reasoned that diminished oxidative stress in PrP^C OE cells may relax antioxidative defenses, rendering them more susceptible to strong oxidative bursts like ferroptosis.

To test this, we treated PrP^C OE cells with a specific inhibitor of system xC−, imidazole ketone erastin (IKE), which blocks glutathione uptake (Fig. 2B). Consistent sensitization was observed at all tested concentrations but was rescued by the addition of aToc, demonstrating specificity for ferroptosis. A simultaneous knockout of GPX8

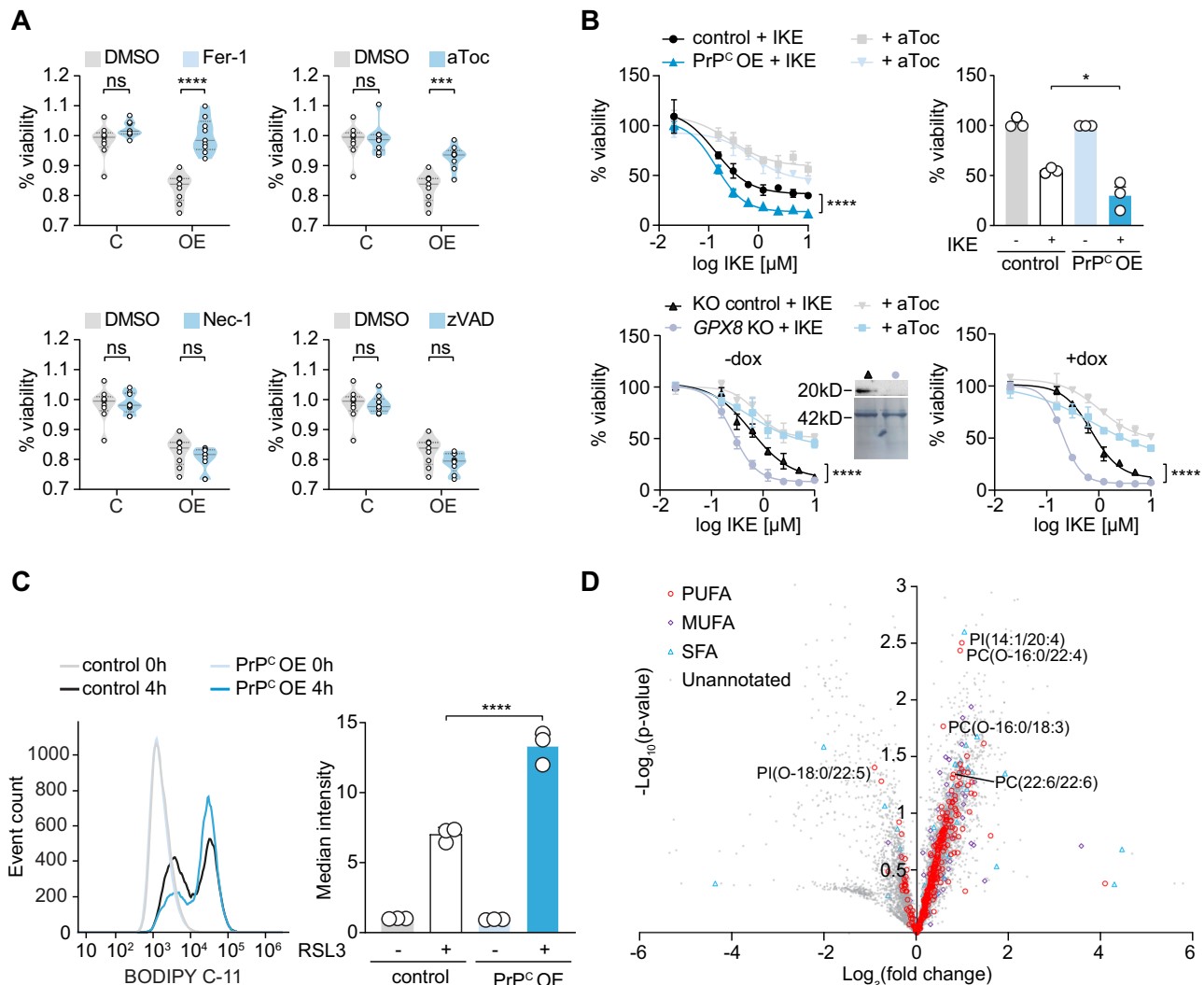

**Fig. 2 | PrP^C sensitizes cells to ferroptosis via lipid remodeling. A** Cell survival of HT-1080 PrP^C OE (OE) and empty vector control (C) cells treated with cell death inhibitors. Cells were infected for 3 days, selected with puromycin for 24 h, then cultured with inhibitors 10 μM α-tocopherol (aToc) or 2 μM ferrostatin-1 as a ferroptosis inhibitor, 10 μM z-VAD-FMK (zVAD) as an apoptosis inhibitor, and 10 μM necrostatin-1 (Nec-1) as a necroptosis inhibitor, respectively, for 3 days. **B** (Left) Dose-response curves illustrating sensitivity of PrP^C-expressing HT-1080 cells (PrP^C OE) compared to empty vector control cells (control OE) to IKE treatment over 16 h. (Right) Survival of PrP^C OE cells compared to control against ferroptosis inducer 0.3 μM IKE (16 h). (Lower panels) Sensitivity of GPX8 knockout (KO) cells with (+dox) and without (-dox) PrP^C OE. Inset shows GPX8 protein levels by Western and total protein loading compared to knockout control cells. Viability data are plotted as representative mean ± SD of $n = 3$ technical replicates for independent experiments repeated at least three times with similar outcomes. **C** PrP^C OE effect on lipid peroxidation induced by 0.1 μM RSL3 induction in HT-1080 OE and control cells measured by BODIPY 581/591 C11 (BODIPY-C11). A typical FACS histogram of $n = 3$ technical replicates of three independent repetitions is depicted. The bar graph shown on the right illustrates the BODIPY-C11 median intensity ± SEM. **D** Comparative analysis of lipid levels in PrP^C OE HT-1080 cells compared to control cells, highlighting fatty acid acyl chain saturation in normalized lipidomics data. Normalization was performed by glucose peak area for each sample and each feature. XCMS data analysis and tentative annotation (of 617 known lipids) were according to the Bruker UPLC-Q-TOF protocol in positive mode, detecting 10,521 total features. Significant lipids have $-\log_{10} P > 1.30$. Viability data are representative means of $n = 9$ (**A**) biological or $n = 3$ (**B**) technical replicates for experiments repeated independently at least three times. $P$-values of two-way ANOVA multiple comparisons with Tukey post-test are shown (**A**–**C**). *$P < 0.05$, **$P < 0.01$, ***$P < 0.001$, ****$P < 0.0001$. Lipid statistics (**D**) were determined by $P$-values ($P < 0.05$, $n = 5$, two-tailed Welch's $t$-test). $n = 5$ PrP^C OE and $n = 5$ control samples. Source data are provided as a Source Data file.

together with PrP^C expression in $PRNP^{tet}$ cells revealed increased sensitivity to IKE, supporting the role of GPX8 in maintaining the redox environment inside these cells. Lipid peroxidation is an essential hallmark of ferroptosis. Application of RSL3 for 4 h resulted in a potent shift in lipid peroxidation in PrP^C OE cells, as detected by the fluorescent membrane peroxidation sensor Bopidy-C11 (Fig. 2C). A substantial portion of PrP^C OE cells die very rapidly from this treatment (Supplementary Fig. 2E) and may only transiently/weakly show Bodipy-positivity. Finally, testing of a panel of anticancer agents moreover showed no discernible nonspecific sensitivity to cell death (Supplementary Fig. 2F). Together, these results demonstrate a specific sensitivity to ferroptosis in PrP^C OE cells.

To further investigate why lower levels of basal oxidative stress lead to ferroptosis sensitivity, we conducted a global lipid mass spectrometry analysis of PrP^C OE compared to control cells. According to lipids determined by total ion chromatogram, significant fold changes were observed for highly unsaturated long-chain containing glycerophospholipids, the primary substrates of lipid peroxidation during ferroptosis (Fig. 2D). Further accumulation of saturated and monounsaturated phospholipids indicated a high level of membrane plasticity that suggested global lipid remodeling. We found that expression of $PRNP$ correlates with $ELOVL1$ and $FADS3$, which respectively confirm the elongation and desaturation of fatty acid precursors (Supplementary Fig. 2G). Interestingly, $ATP2A3$, which increases

calcium concentrations in the ER driving sensitive PUFA-phospholipid formation[27], was found to be strongly inversely correlated, suggesting additional PUFA-lipids in PrP[C] OE are being compensated, to a degree.

Specifically, early ferroptosis lipoxygenase targets of arachidonate and adrenate-containing phospholipids[28] PC(O-38:4) and PI(14:1/20:4) were found to be elevated by 1.91 and 1.96 fold, respectively. Long-chain PUFA-PLs containing two diacyl-PUFA chains PC(22:6/22:6, 1.89 fold) and ether lipids can selectively fuel the spread of lipid peroxidation in a ferroptosis-sensitive environment[29–32]. Thus, we surmise that elevated PrP[C] levels initially limit oxidative stress, allowing for a higher abundance of oxidation-sensitive PUFA-containing phospholipids. However, some cells may exceed this ROS threshold, thus accounting for the protective effect of Fer-1 in untreated PrP[C] cells (Fig. 2A).

## Ferroptosis in bona fide CJD model

Ferroptosis is challenging to detect retrospectively, but several contemporaneous markers are available. One highly specific marker of ferroptosis, fatty acid binding protein five (FABP5), a marker used for retrospective ferroptosis detection in humans[33], was found significantly upregulated in PrP[C] OE cells (Fig. 3A–C). Other proposed markers cyclooxygenase-2 (PTSG2), transferrin receptor (TFRC; by RNA but not Western), heme oxygenase 1 (HMOX1) and the lipid degradation product 4-Hydroxynonenal (4-HNE) were found to be significantly elevated in PrP[C] OE cells (Fig. 3B), signifying the background presence of ferroptosis in cultured cells upon prion overexpression.

Next, we sought to examine if ferroptosis susceptibility is exclusive to cells expressing the native PrP[C] protein or also present in a bona fide infectious prion model. For this, we chose an established iPS-derived organoid method[34–36], inoculated with homogenates from sporadic Creutzfeldt-Jakob disease (sCJD)-infected individuals or normal brain homogenates (NBH), then cultured for 169 days. The pathogenic sCJD prion is detected in the former homogenates as observed by the RT-QuIC assay[37], while HT-1080 PrP[C] OE are classified as non-infectious/non-seeding even in the presence of RSL3 (Supplementary Fig. 3A, B). In contrast to control organoids, sCJD-treated organoids showed pronounced FABP5 staining that was concentrated in areas of evident cell death as partially indicated by empty or "ghost" cellular structures devoid of organelles (Fig. 3C, Supplementary Fig. 4A). This snapshot indicated that the ferroptosis marker FABP5 is expressed in higher levels in cells from sCJD-treated organoids that may potentially die by ferroptosis. 4-HNE detection revealed a significant increase in lipid peroxide products in sCJD organoids at 56 dpi, but only mildly increased at 169 dpi, possibly due to loss of these unstable products prior to the time of measurement (Supplementary Fig. 4B), while FABP5 staining may persist after cells have died.

Therefore, to further investigate whether FABP5 increases according to the progression of prion disease, we compared mice intracranially mock inoculated with normal brain homogenates (NBH) or RML scrapie prions for protein levels at 80- (mid-incubation period), 108- (early symptom onset), and 160- (terminal disease onset) days post-infection. In isolated brains from these animals at 80 dpi no significant difference was observed, however, at 108 and 160 dpi a striking increase in FABP5 in RML mice was observed (Fig. 3D).

To further correlate the emergence of ferroptosis marker FABP5 in mice to CJD in humans, we examined sCJD brain homogenates and found a significant increase in cumulative FABP5 (Fig. 3E). Thus, the increase of the FABP5 ferroptosis marker is conserved from HT-1080 PrP[C] OE to RML mice to CJD patients.

We then sought to examine the direct effect of infectious prions on ferroptosis in vitro. We tested iPS-derived organoids by treatment with RSL3, collected medium fractions and measured cell death by lactate dehydrogenase (LDH) release. In this assay, RSL3 treatment was rescued with alpha-tocopherol, demonstrating ferroptosis specificity

up to 80 nM (Fig. 3F). We infected organoids with sCJD homogenates, incubated these for 110 days (a time point where the earliest metabolic changes were observed)[35] and subsequently exposed them to 40 or 80 nM RSL3 for 48 h. The results showed a remarkable susceptibility compared to control NBH cultures (Fig. 3G). In contrast, genetic ablation of PRNP in iPSC-derived organoids showed a significant resistance to ferroptosis induction by RSL3 (Fig. 3H). To exclude the possibility of CJD-treated organoids being nonspecifically susceptible to lethal stimuli, we treated NBH and CJD samples with staurosporine but observed no changes in sensitivity (Supplementary Fig. 4C). The specific susceptibility to ferroptosis was further supported by increased FABP5 marker deposition in wild-type organoids treated with 40 nM RSL3 compared to PRNP KO organoids, particularly in densely neuronal regions as demarcated by MAP2 expression (Supplementary Fig. 5). Finally, to test if neural cells are directly susceptible to ferroptosis, we expressed human PrP[C] or knocked out Prnp in glutamate-sensitive neuronal HT22 cells. We observed the same effect as in HT-1080: HT22 PrP[C] OE significantly sensitizes cells to RSL3, while Prnp KO has no effect on viability (Supplementary Fig. 6A). Together, these results show that both PrP[C] and infectious prions drive ferroptosis sensitivity.

## Synergistic vulnerability with RAC3

In HT-1080 PrP[C] OE, sCJD-infected organoids and HT22 cells, we observed a fraction of cells that were highly responsive to ferroptosis induction. This suggested that a more complex mechanism or local milieu could influence cellular vulnerability. We therefore chose to identify factors that sensitize to PrP[C] by conducting a whole-genome CRISPR-activation[38] screen in doxycycline-inducible PRNP[tet] cells. The inducible system enables transduction with the sgRNA library prior to PrP[C] overexpression and ensures that control cells have the same complement of guides before ferroptosis induction.

We infected cells with the guide library 3 days prior to doxycycline treatment, split the culture, then supplemented media with a low dosage of ferroptosis inducer IKE (500 nM) for 16 h to induce synthetic lethality (Fig. 4A). Next, we isolated surviving cell genomic DNA for PCR amplification and guide sequencing, and compared frequencies in the control population versus PrP[C]-overexpressing cells, in which depleted guides are expected to correlate to genes promoting PrP[C]-sensitized ferroptosis.

A clear pattern emerged with significantly depleted guides in the surviving PRNP[doxi] population corresponding to RAC3 (p = 0.0007, two-tailed t-test) and its partner RHOC (p = 0.010) small GTP-ases[39], while RHO-interactors[40] EXOC3 (p = 0.0267) and EPHB4 (p = 0.0678) were enriched (Fig. 4B). A highly depleted hit, glycosyltransferase ABO (p = 0.001), was recently implicated in Covid pathogenesis via its glycosylation activity. In contrast, SNX7, a factor proposed to assist autophagosome lipid degradation and the transferrin receptor[41], was significantly enriched (p = 0.0015). Given the evidence of Rho-GTPase involvement, we focused on the most depleted gene, RAC3, a member of the RAS family of small GTPases.

Similar to GPX8, high Rho/RAC family expression is strongly associated with cancer mesenchymalization, metastasis, and stemness[42–44]. Ferroptosis sensitivity has been strongly linked to a high mesenchymal state[7]. Thus, if RAC3 facilitates PrP[C]-induced ferroptosis, we hypothesized it would be hazardous for cells to simultaneously express both genes. Indeed, we observed a significant decrease in RAC3 RNA and protein in PrP[C] OE cells, while ectopic expression of RAC3 also diminished PrP[C] expression (Fig. 4C). In light of this relationship, we investigated their nearest gene neighborhood by CRISPR gene effect analysis (Fig. 4D). Profiling of 17,931 genes and their survival scores in 1095 whole-genome screens by principal component analysis revealed a striking 2D proximity of RAC3 and GPX8, strongly suggesting a shared role in cellular processes and paralleling the observed expression correlation of GPX8 and PRNP (Fig. 1D).

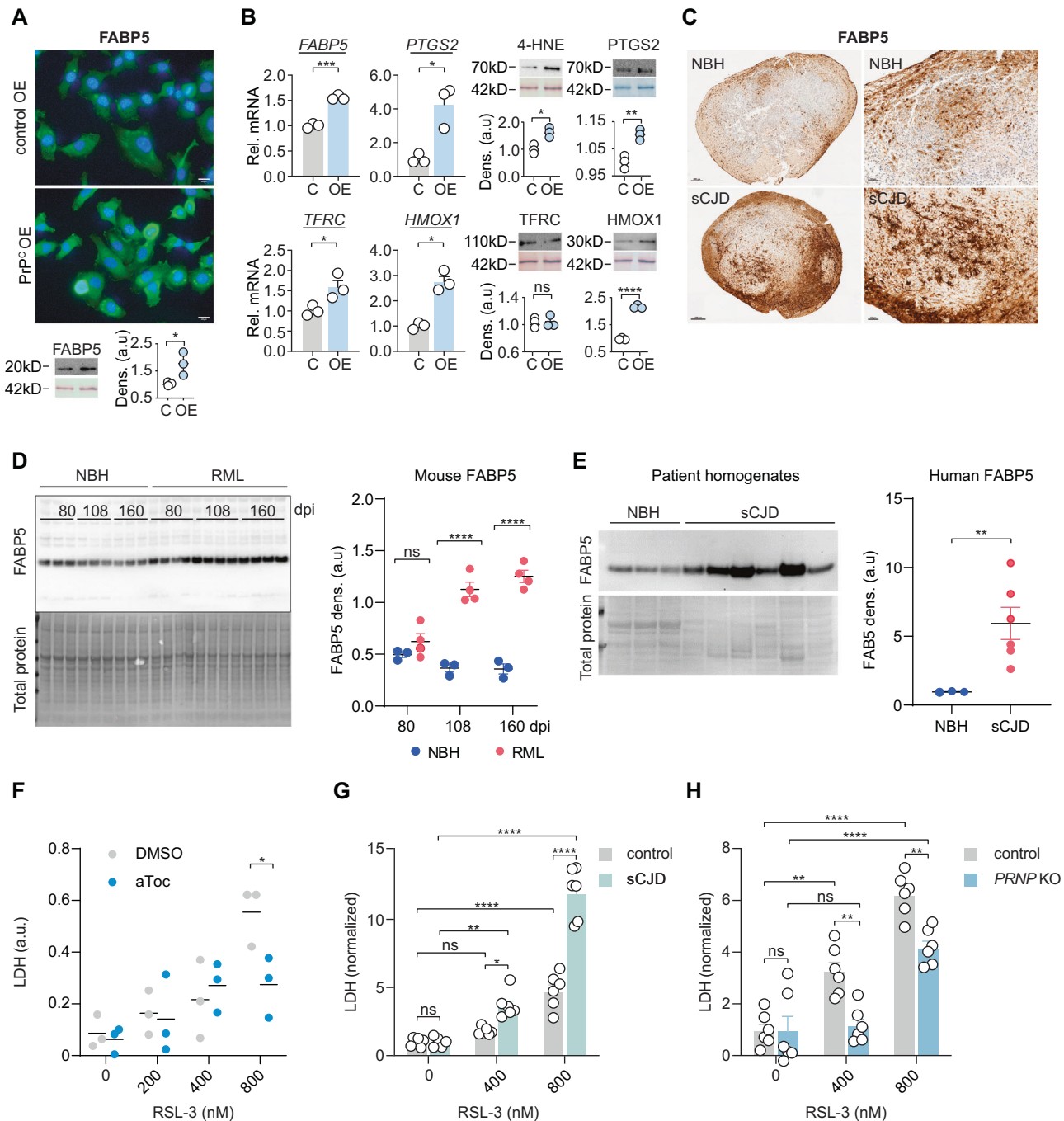

**Fig. 3 | Native and infectious CJD prions drive ferroptosis sensitivity. A** FABP5 changes in PrP$^C$ OE and empty vector control cells by immunostaining and Western with densitometry. Pictures shown are representative results of three independent repetitions performed in triplicate with similar outcomes (scale bar = 20 μm). **B** Relative expression of ferroptosis markers in PrP$^C$ OE (OE) and control (C) cells detected by qPCR and Western. Lower blots display total protein loading. **C** Comparison of FABP5 staining intensity in iPS-derived brain organoids infected with normal brain homogenates (NBH) and sCJD prion homogenates (169 days post-infection, dpi). Pictures shown are representative results of at least three independent repetitions performed with similar outcomes of minimum 10 organoids (scale bar = 200 μm (left panels) and 50 μm (right panels)). **D** Western blotting and densitometry comparing FABP5 protein levels in the brain tissue of mice inoculated with normal brain homogenate (NBH; control) or RML scrapie brain homogenate and sacrificed at 80, 108, and 160 days post inoculation (dpi). **E** Western blotting and densitometry comparing FABP5 protein levels in the brain tissue of people who died from sporadic CJD with tissue from patients who died of a

non-brain-related condition. **F** Lactate dehydrogenase (LDH) release level of brain organoids treated with increasing RSL3 concentrations and aToc rescue for ferroptosis specificity. **G** Normalized survival of brain organoids infected with sCJD (MM1) prion homogenates for 90 days compared to NBH controls against an RSL3 treatment at indicated concentrations (scale bar = 200 μm). Cell death was measured by LDH release activity (48 h, relative to DMSO). **H** Normalized survival of *PRNP* KO compared to NBH controls treated with RSL3 and measured by LDH release (48 h, relative to DMSO). Western data (**A**, **B**) is shown as mean ± SD of *n* = 3 technical replicates. Relative mRNA expression (**B**) is shown as mean ± SEM of *n* = 3 technical replicates. LDH release data and Western data (**E**) are plotted as representative mean ± SEM of *n* = 3 (**D**, **F**) or *n* = 6 (**E**, **G**, **H**) biological replicates for independent experiments. Significance was determined by two-tailed *t*-test (**A**, **B**, **F**), two-tailed Welch's *t*-test (**E**) and two-way ANOVA comparisons, Tukey post-test (**D**, **G**, **H**). *$P < 0.05$, **$P < 0.01$, ***$P < 0.001$, ****$P < 0.0001$. Source data are provided as a Source Data file.

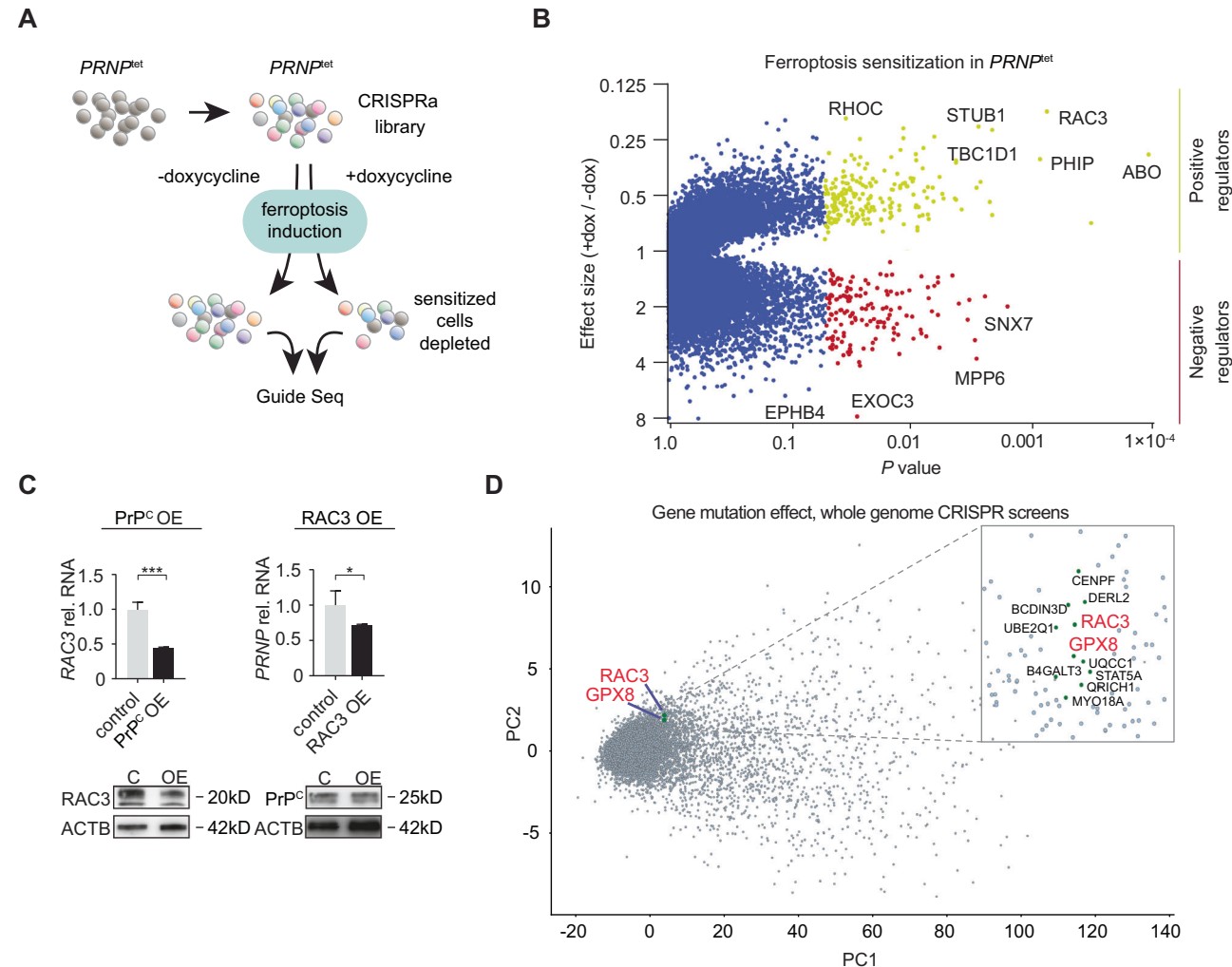

**Fig. 4 | A CRISPR-activation screen identifies RAC3 in prion-induced ferroptosis. A** Schematic of a whole-genome CRISPR-activation depletion screen. HT-1080 conditionally expressing PrP^C (*PRNP*^tet) were transduced with a synergistic activation mediator pooled human library and dCas9-VP64 lentiviruses. PrP^C expression was induced for 3 days with doxycycline or vehicle, followed by treatment with 500 nM IKE, resulting in a viability loss of 5–10%. Normalized guide frequencies amplified from viable cells in sensitized PrP^C cells were scored against (-)doxycycline control cells to identify enriched or depleted guides. **B** Fishtail plot of prion-facilitated ferroptosis by differential CRISPRa guide representation. Effect size is determined by the average *PRNP*^tet (+)doxycycline / (−)doxycycline ratio. Depleted guides (Effect size <1) are positive regulators of PrP^C-facilitated ferroptosis. **C** Inverse relationship of RAC3 and PRNP in PrP^C OE or RAC3 OE, respectively, by qPCR and Western blot. Relative mRNA expression is shown as mean ± SD of *n* = 3 technical replicates. **D** Gene effect characterized by principal component analysis. CRISPR gene effect (individual gene knockout effect on viability and growth, scaled for whole-genome libraries) in 1095 cell lines is displayed in two dimensions. The distance between two genes in the scatter plot can be interpreted to have similar or inverse effects on cellular processes across individual cell lines, relative to all other cell lines. Significance was determined by two-tailed *t*-test. *$P < 0.05$, **$P < 0.01$, ***$P < 0.001$, ****$P < 0.0001$. Source data are provided as a Source Data file.

## RAC3 expression sensitizes cells to PrP^C-induced ferroptosis

Ferroptosis inducers like RLS3 were identified by inducing cell death in KRAS-expressing lines. We tested whether RAC3 increases ferroptosis vulnerability. We tested the inhibitor EHop-016, which inhibits RAC activity and RAC-directed membrane effects via guanine-exchange activity inhibition. Stimulation of ferroptosis with IKE in HT-1080 cells was partially blocked via co-treatment with EHop-016 compared to vehicle, while elevated *RAC3* expression, in contrast, sensitized to ferroptosis induction (Fig. 5A, B). We then sought to test whether combined overexpression of *PRNP* and *RAC3* potentiated ferroptosis sensitivity, but were unable to maintain such clones.

We therefore compared HT-1080 cells freshly infected with either a PrP^C or *RAC3*-containing construct and an empty overexpression vector to doubly infected PrP^C/RAC3 cells (Fig. 5C). In all instances, treatment with Ferrostatin-1 significantly rescued viability. We also tested in parental cells whether *RAC3* knockdown abrogated cell death and observed a significant increase in survival (Supplementary Fig. 6B).

RAC GTPases can activate NADPH oxidases to produce ROS[45]. We tested cytosolic ROS levels in *RAC3* OE with DCFH-DA. While unstimulated cells show no difference from parental controls, stimulation with RSL3 for 3 h revealed a sharp increase in oxidized signal marker (40.8% vs 22.2.%, Fig. 5D), demonstrating potent ROS induction. This revealed a contrasting phenotype to PrP^C OE cells, which suppress cytosolic ROS prior to RSL3 treatment (Fig. 1C). A comprehensive metabolite analysis revealed a clear separation of *RAC3* OE and control cells by principal component analysis (Fig. 5E), indicating that a biochemical restructuring has already occurred following increased expression. Interestingly, *RAC3* OE cells were separated on the second component axis to a degree comparable to IKE treatment of control cells, suggesting they affect many metabolites similarly. However, treatment of control cells with IKE increased PC 1 differences, while treatment of *RAC3* OE only triggered minimal changes. These differences may be due to the long (14 day) duration of *RAC3* overexpression vs short (16 h) IKE treatment. These results suggest that *RAC3* OE

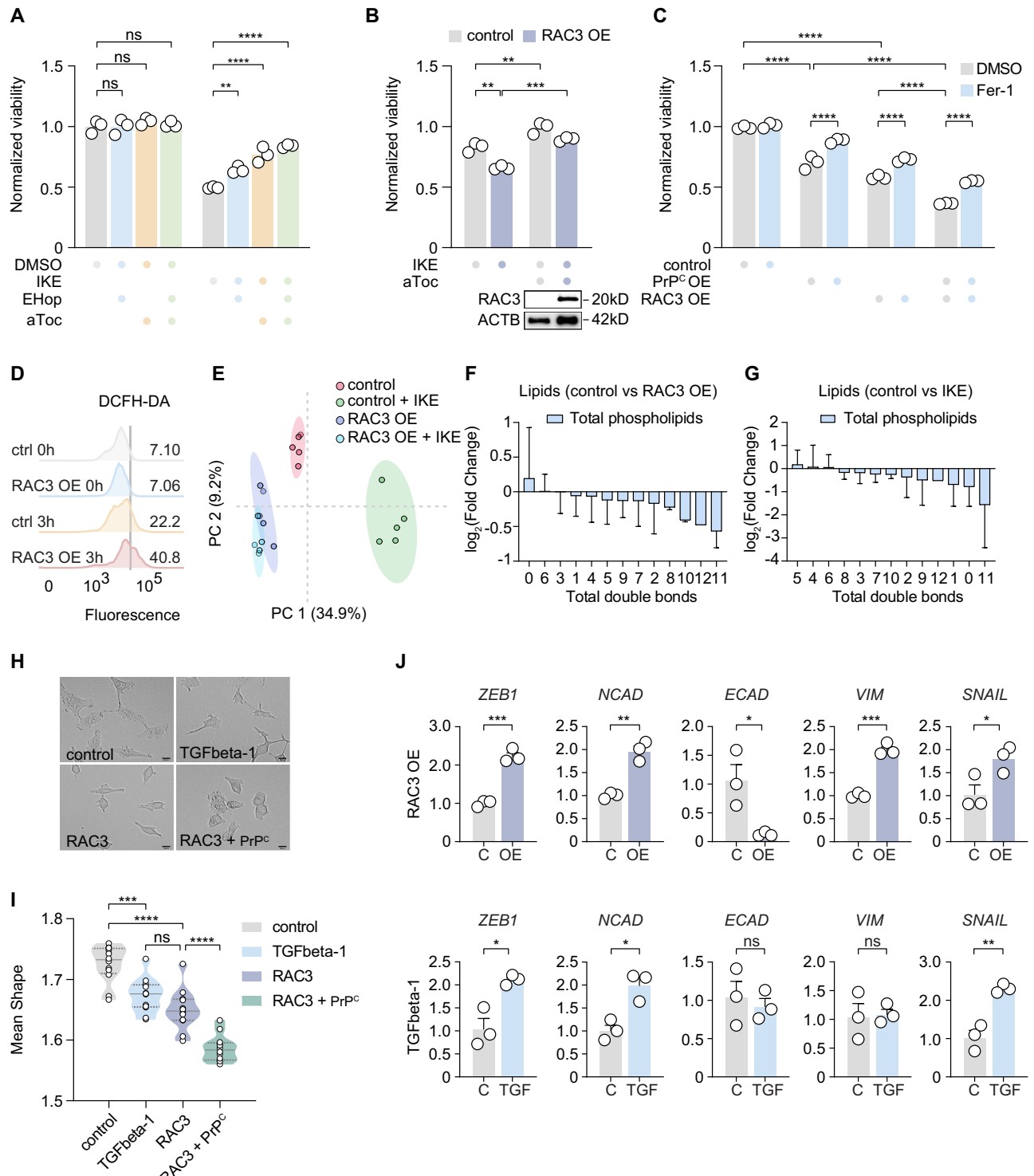

induces a partial metabolic signature, similar to IKE-treated cells. If this hypothesis were valid, metabolites, such as lipids, would be similarly affected in susceptible cells.

To test this hypothesis, we analyzed total ion chromatograms of 210 annotated PUFA and ferroptosis-resistant MUFA-containing phospholipids. We observed a correspondent distribution in *RAC3* OE and IKE-treated cells, suggesting that RAC3 functions similarly to deplete ferroptosis targets of PUFA-glyceryophospholipids (Fig. 5F, G). If these lipids were degraded, replenishment would likely require de novo lipid synthesis or scavenging. We observed correspondent expression of *RAC3* and fatty acid synthase (*FASN*) by linear regression

analysis (Pearson $R = 0.465$), suggesting cellular restocking of lipids occurs partially via de novo lipogenesis (Supplementary Fig. 6C).

One notable feature of HT-1080 *RAC3* OE cells is a clear loss of adhesion along with cellular rounding, a phenotype typically associated with mesenchymalization (Fig. 5H). We quantified this rounding using length-versus-width parameters and compared it to treatment with soluble TGFbeta-1 protein, which induces epithelial-to-mesenchymal transition. While *RAC3* OE resulted in roundness comparable to TGFbeta-1, co-expression with *PRNP* potentiated this phenotype significantly (Fig. 5H, I). Further confirmation of partial mesenchymalization was observed in *RAC3* OE by increased

**Fig. 5 | Prion-induced ferroptosis is facilitated by RAC3-driven mesenchymalization. A** Survival of HT-1080 cells treated with RAC-inhibitor EHop-016 (2 µM) and ferroptosis inducer (IKE) rescued by 10 µM α-tocopherol (aToc). **B** Viability bar graph of RAC3 expressing cells (RAC3 OE) compared to empty vector control with ferroptosis inducer 1.25 µM IKE and 10 µM αToc as a ferroptosis inhibitor. A Western indicates increased expression of RAC3 protein. **C** Cell survival of PrP$^C$ OE, RAC3 OE, and RAC3+PrP$^c$ co-expressing cells was detected in freshly infected cell. Cells were treated subsequently with either 2 µM Ferrostatin-1 or DMSO for 3 days. **D** A flow cytometry histogram depicts the effect of RAC3 OE on induced ROS level via 0.3 µM RSL3 treatment in HT-1080 cells for 3 h measured by DCFH-DA after the indicated time. **E** Principal component analysis of all metabolomic and lipidomic mass spectrometry features for RAC3 OE cells and cells treated with IKE, compared to control cells. (PC, principal component). **F** Bar graph illustrates differential expression of 210 annotated phospholipids and lysophospholipids with varying total numbers of double bonds in RAC3 OE cells compared to control cells with an empty vector. Data are representative means ± SD of $n$ = 5 technical replicates. **G** Differential expression of 210 annotated phospholipids and lysophospholipids

with varying total numbers of double bonds in IKE-treated cells compared to vehicle-treated controls. Data are representative means ± SD of $n$ = 5 technical replicates. **H** Brightfield images of control, RAC3 OE, TGFbeta-1 treated, and RAC3+PrP$^C$ OE co-infected cells indicating various levels of roundness (scale bar = 20 µm). **I** Aspect ratio depicting the effects of TGFbeta-1 treatment, RAC3 OE cells, and RAC3+PrP$^C$ co-expressing HT-1080 cells (mean shape, length/width ratio). **J** Relative expression of epithelial and mesenchymal markers in RAC3 OE and TGFbeta-1 treated cells (3 days) compared to empty vector or DMSO-treated control detected by qPCR. Relative mRNA expression is shown as mean ± SEM of $n$ = 3 technical replicates. Viability data (**A–C**) are representative means ± SEM of $n$ = 3 biological replicates for experiments repeated independently at least three times. The aspect ratio (**I**) are representative means of $n$ = 12 replicates measured by high content microscopy. Significance was determined by two-way ANOVA multiple comparisons with Tukey post-test (**A–C**, **I**) or two-tailed $t$-test (**J**). *$P$ < 0.05, **$P$ < 0.01, ***$P$ < 0.001, ****$P$ < 0.0001. Source data are provided as a Source Data file.

transcription of *ZEB1*, N-cadherin, Vimentin, and Snail1, while E-cadherin, a marker for epithelium, was decreased. The observed effect was as potent or stronger than the addition of TGFbeta-1 (Fig. 5J, Supplementary Fig. 6D, E). *PRNP* expression is moreover tightly linked to the TGF-induced gene *TGF1I1* expression as well as Hippo pathway members responsible for communicating TGF signals (Supplementary Fig. 7A). TGF and associated metastatic processes are strongly linked to poor survival; thus, it is also critically noted that high *PRNP* levels also predict poor survival in Kaplan–Meier analyses of colon and liver carcinomas as well as glioblastoma (Supplementary Fig. 7B). Taken together, RAC3 and PrP$^C$ synergize to induce a mesenchymal phenotype which ultimately sensitizes cells to ferroptosis.

## Loss of RAC3 expression in CJD pathology

Since both PrP$^C$ and pathogenic prion can sensitize to ferroptosis, we reasoned that rapidly progressing neurodegeneration and mortality (mean: 4–5 months) in most sporadic CJD patients may produce similar evidence[46]. If this were the case, RAC3 would be expected to decline with the severity of the disease. We analyzed RAC3 protein in infected RML mouse brain homogenates and observed a distinctive and significant loss of RAC3 protein in these samples over time (Fig. 6A). This loss of RAC3 was mirrored in end-stage CJD brains in humans, indicating conservation of this mechanism (Fig. 6B), together with a slight but concurrent increase in GPX8 protein (Supplementary Fig. 8A). We thus investigated the spatial localization of RAC3 in CJD brains. Immunohistochemistry with antibodies against RAC3 revealed widespread loss of signal in the frontal neocortex of sCJD autopsy cases compared to control autopsy cases and in sCJD-treated organoids (Fig. 6C, Supplementary Fig. 8B, C). In the frontal cortex of control cases RAC3 is expressed in the neuropil predominantly showing a fine punctate distribution pattern which suggests a mainly synapse-associated localization, while cell bodies do not contain RAC3 (Fig. 6C). In all analyzed sCJD cases (Table 1) RAC3 expression is dramatically reduced: the synaptic pattern is only faintly recognizable indicating not only a decreased synaptic RAC3 expression but also a loss of synaptic structures.

## Discussion

Prion infection was previously shown to increase lipid peroxidation and sensitize to oxidative stress[47], but viability loss has been under-investigated due to the absence of other characterized cell death mechanisms at the time of discovery. Here, we revisited the mechanism of prion-induced cell death sensitivity in the context of ferroptosis-sensitive cells and infectious prion organoids to uncover key hallmarks of ferroptosis in the form of elevated biomarkers, sensitizing lipids and peroxidation, a ROS imbalance, and a strong linkage to mesenchymalization processes. A disparity in the interpretation of the role of

ROS in prion pathology has been noted in several studies[48,49]. Importantly, these discrepancies likely result from different cellular systems and treatment durations, as ferroptosis-sensitized cells will eventually disappear from culture and therefore are not subjected to further investigation, while cells that develop resistance to ferroptosis are likely to be maintained and studied. For this reason, we focused on acutely expressing prion cultures, specifically those using an inducible system.

We found that ectopic expression of the native prion PrP$^C$ drives expression of the ER-resident protein GPX8, whose primary role is to detoxify $H_2O_2$ resulting from protein (re)folding. This increase in glutathione peroxidase activity results in a low-ROS state that permits accumulation of ferroptosis-sensitive lipids such as the arachidonic acid-containing glycerophospholipid, rendering cells sensitive to a strong oxidative "pulse" and lipid peroxidation. Although in the first instance it is counterintuitive that cells with lower ROS burden are more susceptible to ferroptosis, this concept is supported by the protection of PrP$^C$-OE cells with long-term treatment of Fer-1. It is expected that cells experience a transient increase in ROS during cell metabolism or division, and that cells with lower ROS levels readily succumb to these stresses via ignition of the lipid peroxidation cascade and are therefore protected by Fer-1. Moreover, cells with higher tonic ROS levels must have established an antioxidant system or a more saturated lipid repertoire to cope with these higher levels.

We observed that GPX8 facilitates PrP$^C$ degradation. One interpretation of this might be that the favorable oxidation conditions in the ER allow for efficient proteosome targeting of PrP$^C$. This would be consistent with the body of literature showing PrP$^C$ conformational changes that block degradation[50,51]. Notably, *GPX8* expression is tightly linked to mesenchymal processes. This may be due to the fact that highly motile cells require flexible membranes comprised of PUFA-containing phospholipids. These lipids are, in turn, sensitive to membrane ROS, which thus requires protection via the glutathione peroxidase system. This would account for the PrP$^C$/GPX8 mesenchymalization axis and a dampened oxidative state in the ER, likely leading to an overall lowering of antioxidant systems in the cytosol, which render the membrane sensitive to strong oxidative pulses delivered by ferroptosis inducers, or by exposure to infectious prion particles (Fig. 6D)[52].

Native PrP$^C$ protein is an absolute requirement for prion propagation and disease progression, as shown by the absence of degeneration in *Prnp*-ablated species[53]. However, the role of PrP$^C$ in neuronal toxicity is much less defined. PrP$^C$ levels have been shown to decrease over the course of prion disease[54]. Like RAC3, this may be explained by the selective loss of cells or structures containing high levels of PrP$^C$. Alternatively, if the PrP$^C$ levels are dropping due to consumption as the substrate for conversion into prions and deposition of these within the

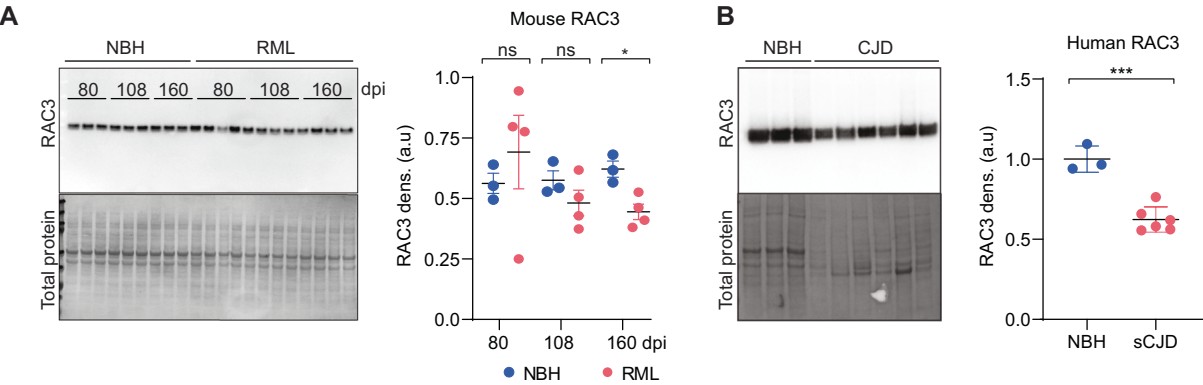

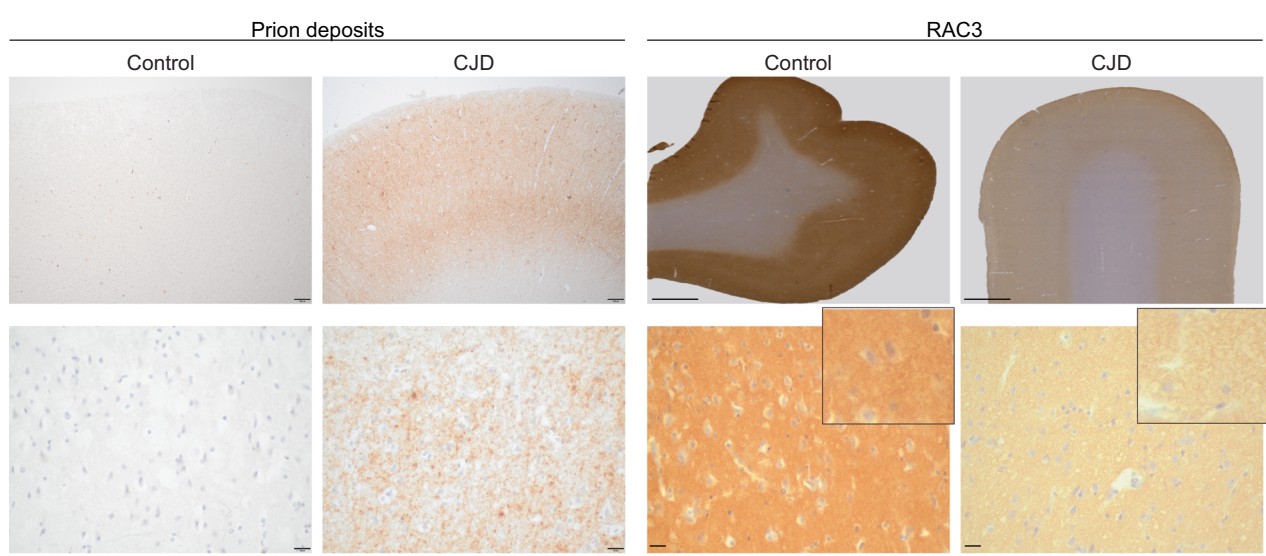

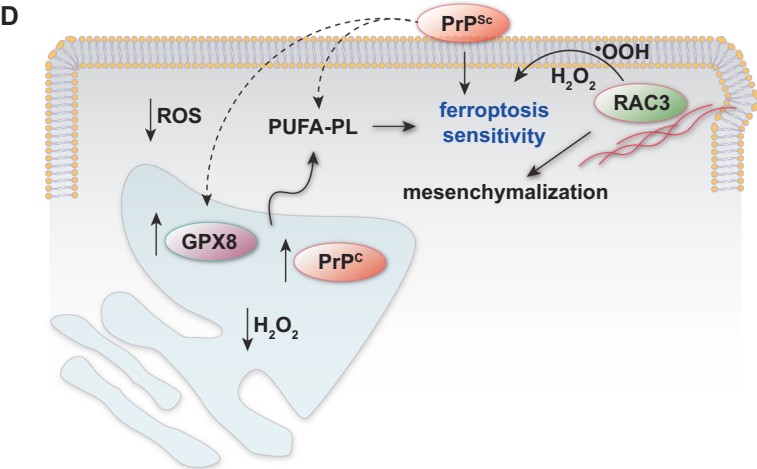

brain, disease subtypes or strains that convert and deposit fastest may demonstrate slower pathogenic decline, due to less functional PrP$^C$ remaining to prime cells for ferroptosis.

Cells displaying a mesenchymal phenotype are exquisitely sensitive to ferroptotic ROS[7]. A high representation of Hippo genes in the YAP1/TAZ pathway was also associated with high PrP$^C$ levels, consistent with a role in mesenchymalization and cancer stem cells found in patient cohorts[8] (Supplementary Fig. 6B). Focal adhesion genes strongly dovetail with *PRNP* expression, further underscoring a change in cellular phenotype. Under nonpathological conditions, this

phenotype may persist indefinitely. However, insult due to intrinsic phenomena such as inflammation or extrinsic oxidative stress may be sufficient to cause a chain reaction, ultimately resulting in ferroptosis in these sensitized cells. Interestingly, expression of vimentin is highly associated with dendritic damage in Alzheimer disease[55], suggesting that mesenchymalization may be a common misfolding- or damage-induced response in both Alzheimer and CJD neurons.

RAC is reported to drive epithelial-to-mesenchymal transition in a ROS-induced feedback loop via activation of the NADPH oxidase (NOX) complex[56]. NOXes are responsible for the production of

**Fig. 6 | RAC3 expression in RML mice and CJD patient brains. A** Western blotting and densitometry comparing RAC3 protein levels in the brain tissue of mice inoculated with normal brain homogenate (NBH; control) or RML scrapie brain homogenate and sacrificed at 80, 108, and 160 days post inoculation (dpi). Total protein is shown in the lower blot. Data are representative means ± SEM of $n = 3$ (NBH) or $n = 4$ (RML) biological replicates. **B** Western blotting and densitometry comparing RAC3 protein levels in the brain tissue of people who died from sporadic CJD with tissue from patients who died of a non-brain-related condition. Total protein is shown in the lower blot. Data are representative means ± SEM of $n = 3$ (NBH) or $n = 6$ (sCJD) biological replicates. **C** (Left panels) Immunohistochemical detection of prion protein deposits in the cortex of the superior frontal gyrus of Creutzfeldt-Jakob disease (CJD) case (left column) and control case (upper panels, scale bar = 20 μm). A higher magnification shows a punctate synaptic distribution pattern of prion protein deposits typical for both sporadic MM/MV1 CJD cases, whereas no deposits are detectable in the control case (lower panels; scale bar = 200 μm). (Right panels) Immunohistochemical detection of RAC3 in the superior frontal gyrus in a control case (left column) and a case with Creutzfeldt-Jakob disease (CJD, right column). RAC3 is strongly expressed in the neocortical neuropil

of the control case (upper left; scale bar = 2 mm), showing a punctate synaptic distribution pattern at higher magnification without expression in cell bodies (lower left; scale bar = 20 μm). In the CJD case, total immunohistochemical signal for RAC3 is strongly reduced in the overview image (upper right; scale bar = 2 mm) compared to the control case. The synaptic distribution pattern is diffuse and unrecognizable at higher magnification (lower right; scale bar = 20 μm). The baseline characteristics of the 6 enrolled patients (3 neuropathological cases and 3 controls) are listed in Table 1. **D** A model of prion-induced ferroptosis sensitivity. PrP$^C$ expression lowers ER ROS levels via increased GPX8 activity, leading to a relaxed oxidative environment that favors the accumulation of ferroptosis-sensitive PUFA-PL. Oxidative stress facilitated by RAC3 simultaneously depletes these lipids and triggers mesenchymalization, sensitizing to ferroptosis. PrP$^{Sc}$ is speculated to contribute to ferroptosis by increased lipid peroxidation, consistent with observations in infected mice, or by cellular sensing of misfolded PrP, resulting in GPX8 upregulation and sensitivity in the presence of RAC3. Significance was determined two-tailed $t$-test (**A**, **B**), $P$-values = 0.0135 (**A**) and 0.0003 (**B**), respectively. Source data are provided as a Source Data file.

## Table 1 | Neuropathological cases and brain sections analyzed in this study

| Case number | Sex | Diagnosis | Age at death | Postmortem delay in hours |
|---|---|---|---|---|
| CJD-1 | Male | CJD | 65–70 | Approx. 34 |
| CJD-2 | Male | CJD | 70–75 | 75 |
| CJD-3 | Male | CJD | 70–75 | 85 |
| Control-1 | Female | n.a. | 75–80 | 89 |
| Control-2 | Male | n.a. | 65–70 | 76 |
| Control-3 | Female | n.a. | 70–75 | 36–46 |

hydrogen peroxide and ROS at the membrane where ferroptosis is initiated[57,58]. Particularly interesting are the differences in cytosolic ROS measured by DCFH-DA in PrP$^C$ and RAC3 overexpressing cells. While PrP$^C$ OE appears to depress cytosolic ROS levels in response to RSL3, RAC3 reverses this phenotype. Hence, the favorable environment for PUFA-phospholipids becomes toxic when RAC3 is present. In this respect, it is striking that RAC3 expression is depleted in sCJD-infected organoids, RML mice, and CJD patient brains. However, both the organoids and the patients may be classified as "end-stage," after the majority of cell death has taken place. This would imply that cells with higher levels of RAC3 have already succumbed to prion-induced cell death, and that remaining cells express lower RAC3. Interestingly, RAC3 has been found to protect against ferroptosis in ES cells[59]; thus, the context of RAC3 and particularly the proportion in the active GTP-bound state or inactive GDP-bound are likely critical in determining its function and localization. Synaptic loss is another key feature of CJD. Thus, it is tempting to speculate that punctate RAC3 immunopositivity is directly involved in synaptic degradation linked to cell death.

Early studies of neuronal death in prion diseases focused on apoptosis as the prominent mechanism[60]; however, other methods have since been implicated[5]. These include necroptosis[61] and autophagy[62]. Although both autophagy and necroptosis pathways exhibit some cross-talk with mesenchymalization and ferroptosis[63–65], the latter has not been investigated as a mechanism of prion-induced cell death. In addition to our current data, previous studies also indicate the involvement of ferroptosis. For example, the early increase in lipid peroxidation seen in mice accompanying misfolded prion accumulation[10] and analysis of lipid profiles in terminal CJD brain tissue showing a decrease in fatty acid oxidizability[66] support a shift toward neuronal vulnerability to ferroptosis during disease. Even earlier investigations hinted at the possibility of mesenchymal transition, with transgenic *PRNP* OE mice demonstrating heightened susceptibility to oxidative stress insults and increased glutathione activity[67]. The

mechanism by which PrP$^C$ influences mesenchymalization is not fully understood; however, its expression is increased during EMT more than tenfold to enhance glycosylation of the NCAM1 adhesion molecule[68]. Furthermore, PrP$^C$ has recently been linked with a mesenchymal phenotype and poor prognosis in colorectal cancer[69]. How PrP$^{Sc}$ specifically contributes to ferroptosis is difficult to decipher, but possibilities include direct effects on lipid peroxidation[47], retrograde transport[70] and cellular detection of misfolding that contributes to GPX8 activity increase[71,72], or an early compensatory increase in PrP$^{C}$[73] that drives these processes. Certainly, the presence of RAC3 is detrimental, yet it remains an open question whether these processes are directly linked or if the cumulative effect on lipid peroxidation is sufficient to drive lethality.

In conclusion, this study provides insights into the complex mechanisms underlying prion-induced cell death sensitivity. We show that the native PrP$^C$ is responsible for maintaining a low-ROS state through GPX8 activity. The state is favorable for the accumulation of PUFA-phospholipids; however, the state is highly susceptible to ferroptotic ROS, driving sensitivity. Additionally, the results highlight the impact of RAC3, which promotes ROS and disrupts the favorable cellular environment for PUFA-phospholipids in PrP$^C$ OE cells. The loss of RAC3-positive structures in RML mice and Creutzfeldt-Jakob patients strongly supports its involvement in prion toxicity. Understanding these processes contributes to our fundamental understanding of prion biology to help guide future therapeutic approaches.

## Methods
### Ethics approval
This retrospective study involving human participants was in accordance with the ethical standards of the institutional and national research committee and with the 1964 Helsinki Declaration and its later amendments or comparable ethical standards. Data analyses were performed on paraffin-embedded tissue samples from archived material at the Edinger Institute. Use of CJD human pathological material was endorsed by the local ethical committee (Goethe-University Medical School/UCT Frankfurt, Ref. no. 19–431). Postmortem analyses on the material were conducted with informed consent from families. Organoids were approved by: Prion protein function in redox homeostasis and associated failure in prion disease, 17-NIAID-00212, 8/3/2017. The human-induced pluripotent stem cells and brain homogenates used in this study were de-identified before being provided to the researchers at the NIH. Thus, the NIH Office of Human Subjects Research Protections (OHSRP) has determined these samples to be exempt from IRB review as per approved project; Prion protein function in redox homeostasis and associated failure in prion disease, 17-NIAID-00212, 8/3/2017.

All animal experiments were approved by the RML Animal Care and Use Committee under protocols 2019-043 and 2022-045. All mice were housed at the Rocky Mountain Laboratories (RML) in an AAALAC-accredited facility in compliance with guidelines provided by the Guide for the Care and Use of Laboratory Animals (Institute for Laboratory Animal Research Council) with Dark/light cycle: 12 h/12 h, ambient temperature: $70 \pm 2$ F, humidity: approximately 40%. All experiments were approved by the RML Animal Care and Use Committee under protocols 2019-043 and 2022-045.

## Cell lines and culture conditions

Cell lines used in this study: HT-1080 (ATCC #CCL-121), HEK-293T (ATCC #CRL-3216), HT22 (Sigma #SCC129). Each cell line was grown under recommended conditions DMEM (Thermo Fisher Scientific) supplemented with 10% fetal bovine serum. Additionally, the growth medium for all cell lines was enriched with 1% penicillin-streptomycin (Thermo Fisher Scientific) and 1% L-glutamine (Thermo Fisher Scientific). HT-1080 was maintained with 1% NEAA (non-essential amino acids, Thermo Fisher Scientific).

## PrP$^C$ expression and immunostaining

$3 \times 10^3$ PrP$^C$ OE and control cells were seeded in each well of a 96-well plate overnight, washed two times with PBS and fixed with 4% PFA for 10 min. Cells were blocked in 5% BSA Fraction V, 10% FBS, 0.3% Triton-X in PBS and incubated in primary antibody (Supplementary Reagents Table) overnight at 4 °C, followed by two times PBS wash and secondary antibody. Images were taken on a confocal microscope in individual channels and merged in Adobe Photoshop.

## Cell viability assays

$2 \times 10^3$ cells were plated in 96-well plates and exposed to IKE (See Supplementary Reagents Table) for an overnight incubation, following the specific conditions outlined in the legends. Subsequently, resazurin (Sigma) was introduced to the wells to a final concentration of 50 μM. After an 8-h incubation period, the fluorescence readings were recorded using an Envision 2104 Multilabel plate reader (PerkinElmer) with excitation at 540 nm and emission at 590 nm. For each condition, a minimum of three wells were subject to measurement and averaged; in general, all viability is presented as percentage relative to the respective control as mean ± SEM.

## Generation of overexpression and suppression cell lines

Human cDNA for *PRNP* and *RAC3* containing was amplified from HT-1080 cDNA and cloned into lentiviral expression vector pLVTHM (Addgene #12247) or pLV (Addgene #39481) with IRES puromycin. A human *RAC3* and three *GPX8* knockdown guides (See Supplementary Reagents Table) were cloned independently into pLV hU6-sgRNA hUbC-dCas9-KRAB-T2a-Neo (modified from Addgene #71236). The viruses were packaged using a pantropic lentiviral 2nd-generation system. Control cells were infected with pLVTHM empty or lentiCRISPRv2 empty vectors, respectively. Following 3 days of infection, overexpressing cells were selected by corresponding drug resistance markers. Knockdown cells were collected on specified days after infection. Knockout vectors for *GPX8* and murine *Prnp* were designed and cloned into lentiCRISPRv2 (Addgene #52961), transiently transfected 16 h into recipient cells, selected with 1 μg/mL puromycin, then cultivated until assay timepoints.

## Western blot

For each experimental condition, approximately $5 \times 10^5$ cells were seeded overnight in a 6-well plate. Protein samples were collected using 100 μL of RIPA buffer and subjected to sonication and concentration determination by BCA. Subsequently, the samples were separated on a 15% SDS-PAGE gel and transferred onto PVDF membranes. Membranes were blocked with 5% skim milk in TBS-T for 1 h at room temperature, then incubated overnight at 4 °C with the primary antibody. Afterward, the membranes were washed three times with TBS-T and incubated with HRP-coupled secondary antibodies for 1 h at room temperature. Following three additional washes with TBS-T, the signal detection was carried out using ECL (Bio-Rad) according to the manufacturer's instructions.

For human brain homogenate blotting, 15 μl of ~2% (w/v) brain homogenate was boiled in 1x Bolt LDS sample buffer (Invitrogen) containing 6% (v/v) BME for 5 min, loaded into 10-well Bolt 4–12% Bis-Tris gels (Invitrogen), and resolved at a constant 200 V in 1x MES buffer (Invitrogen) for 35 min. Proteins were transferred to a PVDF membrane (Millipore) using the iBlot 2 transfer system (Invitrogen), which was then blocked in SuperBlock Blocking Buffer (Invitrogen) for 10 min at room temperature. RAC3 (ABCAM; AB124943; 1:1000), FABP5 (Proteintech; Cat#: 12348-1-AP; 1:2000) and GPX8 (Genetex; GTX125992; 1:1000) primary antibodies, followed by anti-rabbit HRP conjugated secondary antibody (Abcam; 1:5000) were detected by ECL SuperSignal West Atto Ultimate Sensitivity Chemiluminescent Substrate (Invitrogen) and imaged with a iBright imaging system (Invitrogen). Total protein was labeled with a Ponceau stain solution (0.1% w/v Ponceaus and 5% v/v acetic acid) and imaged by the iBright imaging system. ImageJ 1.52n was used to measure the densitometry density of the bands as well as the total protein. The levels of test protein were normalized to the total protein, and the average levels between CJG and control groups were compared by Welch's unpaired *t*-test (Two-tailed with 95% confidence interval).

## GPX activity detection

The GPX activity assay (ELA-E-BC-K096-S) was carried out using instructions provided by the manufacturer.

## Cytosolic ROS detection

Cells were seeded in a 6-well plate, and the following day, the medium was replaced with 2 mL of medium containing RSL3 at a concentration of 100 nM for 2 h. 2,7-Dichlorodihydrofluorescein diacetate (DCFH-DA) from Biomol (Cay85155-50) was added to the wells at a final concentration of 25 μM to detect cytosolic ROS. The cells were then incubated for an additional 30 min. After removing the medium, the wells were rinsed twice with 2 mL of and 200 μL of Accutase from Sigma (A6964) was added to each well. The detached cells were resuspended in 800 μL of PBS per well and analyzed using an Attune acoustic flow cytometer from Applied Biosystems. Histogram intensities were collected from the BL-1 channel excited by a 488 nm laser. The median fluorescence intensity of each well was determined and normalized to the DMSO-treated control cells using FlowJo 10 software.

## Lipid peroxidation detection

Flow cytometry was employed to detect lipid peroxides. In each well of a 6-well plate, $1 \times 10^5$ cells were seeded. The following day, the medium was replaced with 2 mL of medium containing the ferroptosis inducer RSL3 at a concentration of 100 nM for 4 h. Afterward, BODIPY 581/591 C11 (BODIPY-C11, Thermo Fisher Scientific) was added to the wells at a final concentration of 2 μM to detect lipid peroxidation. The cells were then incubated for an additional 30 min. Following the removal of the medium, the wells were rinsed twice with 2 mL of PBS. To detach the cells, 200 μL of Accutase (Sigma) (A6964) was added to each well. Detached cells were subsequently resuspended in 800 μL of PBS per well and analyzed using an Attune acoustic flow cytometer (Applied Biosystems). Histogram intensities were collected from the BL-1 channel excited by a 488 nm laser. The median fluorescence intensity of each well was determined and normalized to the DMSO-treated control cells using FlowJo 10 software.

## Ferroptosis assay

A total of $4 \times 10^4$ cells were seeded in a 6-well plate for infection. Following a 3-day infection period, cells were selected using

puromycin overnight for 16 h. Afterward, inhibitors were added to the respective wells, and a 3-day co-culture with the inhibitors was carried out. Following trypsinization and counting, an empirically determined volume of control cells containing $2 \times 10^3$ cells was plated in 96-well plates, with the same volume of each experimental condition being added to each well. Resazurin (Sigma) was introduced to the wells (final concentration 50 μM) for 8 h. Fluorescence readings were taken using an Envision 2104 Multilabel plate reader from PerkinElmer (excitation, 540 nm; emission, 590 nm). For each experiment, a minimum of three wells were measured for each experimental condition. The viability data were presented as a percentage relative to the control and expressed as a mean ± SEM.

### High content assay

For ferroptosis biomarker detection, $5 \times 10^3$ cells for each experimental condition were seeded in 96-well plates and allowed to incubate overnight. Immunofluorescence intensity assays were conducted using the Operetta imaging system from PerkinElmer. During the assay, DAPI staining was utilized to identify primary objects, and a Cy3-labeled secondary antibody from Dianova was employed for intensity measurements. The secondary antibody was diluted at a ratio of 1:500. Antibodies used were listed in the Supplementary Reagents Table.

For cell shape, cells were stained with FITC-conjugated phalloidin and imaged in Thermo Scientific CX-7. The parameter "Object-ShapeLWRCh1" was used to measure cell shape based on the ratio of length to width using object-aligned boundary boxes.

### Organoid generation, culture and infection

The undirected organoids used for the current study were differentiated from ACS-1023 (ATCC) or RAH019A[74] induced pluripotent stem cells, maintained in long-term culture and infected with 0.1% (tissue wet weight/volume) sCJD prions (ACS-1023 with MV1 and RAH019A with MM1) from postmortem brain homogenate as described previously[35]. Organoids were infected at 5 months post beginning differentiation and harvested when substantially infected at 179 dpi or treated with RSL3 at 90 days post-infection; the disease timeframe associated with the beginning of metabolic changes[35]. *PRNP* KO organoids were differentiated from RAH019A cells that had been engineered with a frameshift mutation to disrupt the *PRNP* open reading frame as described by Groveman and Williams[75]. Four-five-month-old RAH019A WT and *PRNP* KO organoids were used for RSL3 treatments. Media lactate dehydrogenase (LDH) release assays were carried out using the Cytotoxicity Detection Kit Plus (LDH) from Roche as per the manufacturer's instructions. For histological analysis, organoids were fixed in neutral buffered formalin for approximately 24 h before embedding in paraffin. Two- or five-micron sections were cut for staining as described below.

### RT-QuIC

Cells were cultured − or + doxycycline for 3 days and then seeded at 130,000 cells/well in 6-well culture plates. The next day, cells were harvested by trypsinization and washed with culture medium. Three replicate wells were pooled together to make 4 final replicates of each condition. The pooled cells were centrifuged for 5 min 1500 RPM. The media was aspirated, and the pellets were snap frozen in liquid nitrogen and then stored at −80 °C. Naïve or 22 L infected N2a cells were used as negative and positive controls, respectively.

For RT-QuIC, the cell pellets were resuspended in 250 μL of 0.1% SDS in PBS and briefly sonicated. Lysates were centrifuged at 2000×*g* for 2 min, and the supernatants were recovered and diluted 1:10 or 1:100 in 0.1% (w/v) SDS in PBS with 1x N2. RT-QuIC was performed as previously described with some modifications[76]. Black 384-well assay plates with a clear bottoms (Nunc) were pre-loaded with 49 μL of RT-QuIC reaction mix containing final concentrations of 10 mM sodium phosphate (pH 7.4), 300 mM NaCl, 10 μM thioflavin T (ThT), 1 mM

ethylenediaminetetraacetic acid tetrasodium salt (EDTA), and 0.1 mg/mL hamster recombinant PrP 90–231 (purified as described in ref. 77). Eight replicate reaction wells were seeded with 1 μL of diluted lysate for each sample, also contributing 0.002% SDS to the final reaction mix. Plates were sealed (Nalgene Nunc International sealer) and then incubated in a BMG FLUOstar Omega plate reader at 50 °C for 50 h with cycles of 60 s of shaking (700 rpm, double-orbital) and 60 s of rest throughout the incubation. ThT fluorescence measurements (excitation, 450 ± 10 nm; emission, 480 ± 10 nm [bottom read]) were taken every 45 min. Wells that showed an increase in ThT fluorescence >10% of the maximum baseline-subtracted ThT fluorescence value on the experimental plate were considered positive reactions. A sample was considered positive for prion seeding activity if >25% of the replicate reaction wells had positive responses.

### Lipidomics

See schematic overview (Supplementary Fig. 9).

**RAC3 overexpressing lipidomics study.** A total number of samples was fifteen in three groups. There were five technical replicates for each group (5 samples of the parental HT-1080 cells (control), 5 samples of IKE-treated samples, 5 samples of the RAC3 OE cells). Lipid extraction was performed using the following procedure. Initially, cell pellets were collected in Eppendorf Safe-Lock tubes and treated with 400 μL of methanol cooled on dry ice. The tubes were then vortexed to ensure thorough mixing. Subsequently, the pellets were transferred to MACHEREY-NAGEL GmbH & Co. KG bead tubes (MN Bead Tubes Type A). Any remaining pellet residue was washed with an additional 400 μL of methanol cooled on dry ice, and the mixture was vortexed for 30 s before being transferred to the bead tubes. The pellets were then homogenized in the bead tubes using a homogenizer (Precellys, Bertin Technologies) at 4 °C using the soft mode. After homogenization, the bead tubes were centrifuged at 19,000×*g* for 15 min at 4 °C. Following centrifugation, 400 μL of the supernatant was transferred to a vial. Finally, the supernatant was injected into an LC−MS/MS system, which consisted of a liquid chromatography system (Exion LC series UHPLC, Sciex) and a time-of-flight mass spectrometer (X500QTOF, Sciex). A reverse-phase chromatography column (Cortecs UPLC C18 column, 150 mm × 2.1 mm ID, 1.6 μm, Waters Corporation) was used for lipid separation. The eluents A and B were composed of 60% $CH_3CN$, 40% $H_2O$, 1% HCOOH, 1% $HCOONH_4$ (10 mM) and 90% 2-Propanol, 10% $CH_3CN$, 1% HCOOH, 1% $HCOONH_4$ (10 mM), respectively. A gradient elution method was employed for lipid analysis with a total duration of 25 min and a flow rate of 0.25 mL/min. The gradient program followed the schedule below: 0 min, 68.0% A; 1.50 min, 68.0% A; 21 min, 3.0% A; 25 min, 3.0% A.

Mass spectrometry parameters were set in the electrospray ionization positive mode, with a TOF start mass of 100 Da and a TOF stop mass of 1700 Da. The MS experiment was conducted in Information-dependent acquisition mode (Data-dependent acquisition mode). Raw data was converted into mzXML format using MSConverter (Proteo-Wizard, Palo Alto, CA) in peak picking mode to obtain centroid data. Feature detection, processing, alignment, and identification were performed using MZmine software[78]. For tentative annotation of lipids, the MoNA library was utilized. Lipid differential expression and lipid characteristics were analyzed using LipidSig[79]. PCA analysis was conducted in Metaboanalyst online, based on MZmine data processing[80]. Ellipses show 95% confidence region.

**PrPC overexpressing lipidomics study.** A total number of samples was ten in two groups. There were five technical replicates for each group (5 samples of parental HT-1080 cells (control) and 5 samples of PrP[C] overexpressing (PrP[C] OE)). Lipid profiling of PrPC-overexpressing (PrPC OE) cells and control HT-1080 cells was performed using a Bruker maXis II quadrupole time-of-flight (Q-TOF) mass spectrometer

(Bruker Daltonik GmbH, Bremen, Germany) coupled to a Waters ACQUITY UPLC system. Lipid separation was achieved using a 29-min gradient elution on a UPLC column. The gradient program was as follows: 0–1.5 min, 68% solvent A; 21–25 min, 3% solvent A; 29 min, 68% solvent A. The injection volume was 5 µL. Mass spectrometric analysis was conducted in positive ion mode using electrospray ionization (ESI). The mass range scanned was from 100 to 2000 Da in data-dependent acquisition (DDA) mode. Raw data files were converted to mzXML format after recalibration in enhanced quadratic mode using the Bruker Compass Data Analysis Software (Bruker Daltonik GmbH, Bremen, Germany). Feature detection, peak processing, alignment, and tentative lipid annotation were performed using the XCMS online service[81] with the "UPLC / Bruker Q-TOF pos" data analysis protocol.

### Immunohistochemistry on human brain tissue and organoids

Tissue samples of the anterior superior or medial frontal gyrus were cut out of autopsy brains fixed in 4% neutral phosphate-buffered formaldehyde, put into 100% formic acid for 1 h at room temperature (RT), washed under running tap water for 30 min, fixed again for 24–48 h in neutral phosphate-buffered formaldehyde at RT and paraffinized.

Immunohistochemistry for prion protein including counter-staining with hematoxylin was performed on 5 µm paraffin sections in a Bench Mark Ultra (Roche) according to the instructions of the manufacturer using a mouse monoclonal antibody (clone L42; RIDA, catalogue-no. R8005) diluted 1:300 and a diaminobenzidine/peroxidase based detection system (optiView; Roche). For RAC3, Abcam AB129062 was used at 1:300. Sections were pretreated with proteinase (Roche) for 8 min at RT and boiled in citrate buffer (buffer CC1, Roche) for 32 min.

Images of prion protein and RAC3 stains were taken with an Olympus BX23 camera at an Olympus BX45 microscope using cells-Sens Entry software v3.2.

Organoids were immersed in 3.7% neutral buffered formalin for ~ 24 h prior to standard embedding in paraffin. Two or five-micron sections were cut. Organoid histochemistry staining for FABP5 and RAC3 was carried out as above. MAP2 (1:500; Abcam) and GFAP (1:2500, Dako) were detected using ChromoMap DAB (Roche/Ventana #NC1859896) and photographed using Aperio Imagescope software.

### Real-time PCR

$1 \times 10^6$ cells per sample were seeded, and total RNA was isolated using the TRIzol RNA extraction protocol. Subsequently, 2 µg of total RNA was reverse transcribed using the AMV Reverse Transcriptase Kit (NEB, M0277S). Quantitative PCR reactions were carried out on a Light-Cycler480 (Roche) instrument using Power SYBR Green Master Mix (Lager, 5000989). Differences in mRNA levels compared to control were calculated using the ΔΔCp method, with RPL27 or TBP as housekeeper gene. The qPCR primer sequences are listed in the Supplementary Reagents Table.

### Iron and H₂O₂ determination

The cytosolic LIP was determined as described[82]. For each measurement, $4 \times 10^5$ cells were loaded with 5 nM calcein-AM (Molecular Probes) in PBS at 37 °C for 15 min. After washing non-internalized calcein, the cells were transferred to a 1.5 mL ep tube, and the basal calcein fluorescence was recorded (excitation, 488 nm; emission, 517 nm). HyperER endoplasmic reticulum hydrogen peroxide measurement was performed as described[83]. 500 µM Ferric ammonium citrate (FAC) or 100 µM Deferoxamine (DFO) were treated overnight as the positive and negative control.

FerroOrange. For each FerroOrange measurement, $4 \times 10^5$ cells were loaded with 1uM FerroOrange in non FBS medium at 37 °C for 20 min. After washing two times with PBS, the cells were transferred to 1.5 mL ep tube, and the fluorescence was recorded (excitation, 532 nm;

emission, 572 nm). 500 µM Ferric ammonium citrate (FAC) or 100 µM Deferoxamine (DFO) overnight treated condition were measured as the positive and negative control. For Mössbauer measurements, see Supplementary File 1.

### Informatic analyses

All gene expression correlations and gene effect correlations are derived from Cancer Cell Line Encyclopedia data downloaded 22Q1 and 23Q2 from Depmap databases, respectively. Pearson's expression correlation and $P$-values were calculated for display using Python libraries pandas, scipy, statsmodels, sklearn and matplotlib. All scripts are available upon request.

### CRISPR screen

Human Synergistic Activation Mediator (SAM) library (#1000000057, Addgene) was infected together with lenti dCAS-VP64_Blast (#61425, Addgene) and MS2-P65-HSF1_Hygro (#61426, Addgene) into $PRNP^{xtet}$ cells and selected with respective antibiotics. After 3 days selection, PrP$^C$ was induced for 3 days with doxycycline, or vehicle for controls, followed by treatment with 300 nM IKE, resulting in viability loss of 5–10%. Normalized guide frequencies amplified from viable cells in the sensitized PrP$^C$ cells were scored against to (-)doxycycline control cells to identify enriched or depleted guides using ENCoRE[84].

### Animal studies

Prion inoculation and brain homogenate preparation for western blotting analysis.

Six-week-old wild-type C57BL10 mice were intracerebrally inoculated with 30 µL of 1% (w/v) brain homogenate from RML prion strains, which were prepared from stocks previously determined to have a 50% infective dose (ID50) of $2.4 \times 10^4$ RML. Uninfected control mice were 'mock' inoculated with uninfected or normal brain homogenates (NBH). Mice were monitored twice weekly prior to clinical onset and every 1–3 days throughout the clinical phase. Mice were euthanized at 80, 108, 160 dpi, and brains were collected and sliced into 300 µm thick coronal sections. Those slices harboring the hippocampus, thalamus, hypothalamus, and amygdala (areas of known disease pathology) were homogenized in 1 × PBS into 1% w/v brain homogenate for western blot analysis.

Proteins were denatured by boiling in 1 × Bolt LDS sample buffer (Invitrogen) containing 10% Beta-mercaptoethanol for 5 min at 100 °C and resolved into various sizes in 4–12% Bis-Tris gels (Invitrogen). Transfer, blotting and quantification were done as for the human brain samples. The levels of test protein were normalized to the total protein, and the average levels between uninfected and infected groups were compared by Welch's unpaired $t$-test (Two-tailed with 95% confidence interval).

### Sex and gender

Sporadic CJD qualifies as a rare disease with 1 to 2 new cases per million inhabitants per year. sCJD has a 1:1 male-to-female ratio, and thus sex does not appear to be a factor determining susceptibility or progression of the disease. Thus, sex is not considered in the study design, and the pathological material was assigned to gender according to a licensed pathologist during the autopsy. Anonymized reporting of individual sexes does not influence outcome or predictability in the study.

### Reporting summary

Further information on research design is available in the Nature Portfolio Reporting Summary linked to this article.

## Data availability

All data presented in the manuscript are available in the Supplementary files. Repository accessions and links RAC3 overexpressing and

imidazole ketone erastin (IKE) Lipidomics: MassIVE MSV000097840 and PrP$^C$ overexpressing Lipidomics: MassIVE MSV000097841. Source data are provided with this paper.

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

## Acknowledgements

We thank Maika Dunst for expert technical assistance with tissue processing and immunohistochemistry. Human sCJD brain samples were a kind gift from Prof Gianluigi Zanusso (University of Verona) and include 3 each of two sCJD subtypes (MM1 and MV2). We also acknowledge the help of the Innovation Platform for Academicians of Hainan Province. This project was supported by the German Research Foundation (DFG, Project 501860452) to J.S., T.A., and S.M. and DFG (511521600) to J.S., the EnABLE initiative by the state of Hessen to S.M., and Helmholtz Center Munich to J.S., Genetics and Cell Engineering Group. This work was also supported by the Specific Research Fund of the Innovation Platform for Academicians of Hainan Province (Grant Nos. YSPTZX2022011, YSPTZX2025011) to X.J. and J.S.; and the Undergraduates Training Program for Innovation and Entrepreneurship of Hainan Province (X.J.). This work was also funded in part (C.H., B.G., K.W., S.F., B.R., T.T.) by the intramural research program of the National Institutes of Health (NIAID).

## Author contributions

J.S. conceived the study, supervised the research, and wrote the manuscript together with C.H. and T.A. H.P. and S.P. performed all genetic modifications and cellular in vitro assays, CRISPR screening,

interpretation and statistics. M.Q., C.S. and S.F. performed Western and RT-PCR analyses. X.J., C.C. and X.Z. performed informatics analyses. B.G. and K.W. performed all organoid and RT-QuiC assays. K.K. performed Mössbauer analyses. B.V. and C.M. performed the mass spectrometry and lipidomics analysis. B.R. and S.F. performed animal time course infections and analyzed the tissues. T.A. and S.M. collected patient material, performed histology and immunostainings, and evaluated results for the manuscript. T.T. performed IHC analysis on certain human tissue and organoids. All authors read and approved the final manuscript.

## Funding

## Competing interests

The authors declare no competing interests.
