## [Transparent Peer Review file · Nature Communications]

Prion-induced ferroptosis is facilitated by RAC3

Corresponding Author: Dr Joel Schick

Version 0:

Reviewer comments:

Reviewer #1

(Remarks to the Author)

Summary: The authors investigate a cell death pathway previously unattributed to prion disease, ferroptosis. They show that PrPC sensitizes cells to ferroptosis by maintaining cells in a low redox state, which allows for the accumulation of unsaturated long-chain phospholipids. These lipids are susceptible to peroxidation during bursts of oxidative damage, which ultimately leads to ferroptosis. The authors link the expression of PrPC with that of GPX8 and RAC3, which exacerbate this sensitivity. They present correlates of this in CJD infected cerebral organoids, and in the brains of CJD patients.

Major points:

This paper is exceedingly difficult to read, and its overall message is obscure. There is no attempt to connect the findings to a simple, logical model or hypothesis. To suggest that PrPC and PrPSc both drive susceptibility to the same cell death pathway (ferroptosis) is difficult to accept, and comes with a high burden of proof, which is not met. The data supporting a connection between PrPSc and ferroptosis is particularly weak.

PrPSc drives ferroptosis susceptibility:

1. p6 line 146: regarding CJD infected organoids: "... subsequently die of ferroptosis." This statement is not well justified. The data show only an increased staining of FABP5. Can this be quantified? How many organoids were investigated? Staining of CJD patient samples (Figure 6) would be helpful here.

a. Fig 3D: was the increased sensitivity to RSL-3 induced LDH release, and rescue by aToc, statically significant?

b. Fig 3E shows an increase in LDH release upon RSL-3 treatment of CJD infected organoids compared to non-infected organoids. This is all taken to support the conclusion that "infectious prions drive ferroptosis susceptibility". However, an alternative explanation is that these organoids are more susceptible to RSL-3 because they are already under the stress of prion infection, as opposed to a direct result of PrPSc itself. A demonstration of their response to stimuli promoting other forms of cell death would strengthen this conclusion.

2. Figure 6: The reduction of RAC3 staining in the frontal cortex of CJD patients is the only evidence tying these findings to CJD patients, and prions in general, despite this being in the title of the manuscript. Comparing the staining intensity of brain slices is inherently subjective – can a western blot be performed? It would be useful to analyze other proteins, as mentioned above, such as FABP5 and GPX8 to help substantiate these findings.

a. How many organoids were used to look at RAC3? Can this be quantified? I find these data (Sup Fig 7A) unconvincing in their present form.

3. The reduction of the levels of a protein in the brain of an individual who died of a neurodegenerative disease can be explained in many ways. A time series looking at the levels of RAC3, FABP5, GPX3 and PrPSc over the course of disease (organoids, mice, or brain slices) would help demonstrate the specificity of these changes to the development of prion disease.

PrPC drives ferroptosis susceptibility:

The data connecting PrPC to ferroptosis is somewhat more convincing than that for PrP^{Sc}, but the experiments are done in a cell line that is highly susceptible to ferroptosis. Can these observations be recapitulated in other cell lines, or (perhaps more importantly) in neurons? One would expect from these data that a comparison of Prnp knockout neurons vs WT neurons vs PrP overexpressing neurons would show increasing susceptibility to ferroptosis inducing factors like RSL-3 or erastin.

1. Figure 1A: Iron balance is unaffected by the expression of PrPC. This is a surprising result, considering that the rest of the data connects PrPC expression to ferroptosis. This should be discussed in detail. Is this because of the reduction in PrP expression over time? If so, this analysis bears repeating with the DOX inducible cells.
2. The connection between PrP expression and a reduction in cell viability is somewhat casually introduced, with supporting data not displayed until later in the manuscript (Fig 2A). This would benefit from some restructuring. Further, while the use of resazurin shows a reduction in the viability of PrP overexpressing cells, a live/dead assay would better support the statement that these cells are "lost" over time.
3. Figure 1 G-H: The concomitant increase in GPX8 expression along with PrPC, and subsequently increased peroxidase activity and lower H₂O₂ levels could be explored further. For example, does GPX8 knockdown in PrPC overexpressing cells return the peroxidase activity and H₂O₂ levels to those observed in non-PrPC overexpressing cells?
4. Figure 2C: Why don't PrPC overexpressing cells demonstrate higher levels of lipid peroxidation without RSL3 induction? Shouldn't this be the case, given that they have increased staining for FABP5 (Figure 3A)?
 - a. Figure 3A: A western blot for FABP5 would be more convincing. It would also be nice to see confirmation of the qPCR data by western blotting for PSTG2, TFRC, and HMOX1.
5. The data presented in Sup.4B of single organoids (WT vs PRNP KO) treated with the ferroptosis inducer RSL3 are not convincing. Are there additional images, western blots, or quantification that can bolster the conclusion that there is reduced FABP5 staining in PRNP KO organoids?
6. Figure 4C: Are these differences statistically significant? The PrP signal in RAC3 overexpressing cells does not match the quantification. On this note, the western blots for PrP show only a single band – PrP typically migrates as several bands due to differential glycosylation and physiological cleavage events.

Minor points:

p2 line 23: "Although the precise mechanism by which the prion protein kills neurons...". This should read: "... by which prions kill..." the prion protein (PrPC) is not in and of itself toxic to neurons.

p3 line 66: "We observe that cells constitutively expressing PrPC lose expression over time, despite selection with a dicistronic puromycin resistance gene, while PRNP (the gene encoding PrP_C) knockout cells grow faster (Fig. 1B, Supp. Fig. 1C)". It is not clear how these statements are related to one another through experimentation.

p4 line 76: "As an endoplasmic reticulum resident protein, GPX8 detoxifies H₂O₂ generated during disulfide isomerase mediated protein (re-)folding and is tightly associated with mesenchymalization processes. Critically, the prion protein has shown a susceptibility to misfolding." This statement is unnecessary and implies that GPX8 is somehow involved in PrP misfolding, which, as far as I am aware, is not the case.

p5 line 134: The reader is directed to Supp Fig3A regarding FABP5 expression, but this figure shows RT-QulC assays of CJD samples.

p6 line 146: "...subsequently die of ferroptosis.". It has been established that these organoids show higher levels of FABP5, but not that they subsequently died of ferroptosis.

p7 line 174: " A highly depleted hit, glycosyltransferase ABO, was recently implicated in Covid pathogenesis via its glycosylation activity, a process that is known to affect prion localization and function." The reference (44) shows only that PrP is glycosylated, not a connection with this particular glycosyltransferase, or SARS-CoV-2 pathogenesis.

The role of PrP in EMT has been described previously (Mehrabian, et al. "The prion protein controls polysialylation of neural cell adhesion molecule 1 during cellular morphogenesis." PLoS One 10.8 (2015): e0133741). This paper should be discussed.

Reviewer #2

(Remarks to the Author)

Peng et al report on novel MoA of how prion induces degeneration through ferroptosis via reducing ER-ROS involving

GPX8 and RAC3 driven mesenchymalization.

- Intro or discussion. Please include recent pioneering report linking YBX1/RAC3 axis to ferroptosis (10.1038/s41419-024-06882-5)
- Fig. 1: Please include positive and negative ctrl also using FerroOrange. An CE-ICP-MS approach measuring ratios Fe3+/3+ would be ultimate proof and recommended.
- Fig. 1: To strengthen conclusion on involvement of GPX8, one should perform knockdown of GPX8 in PrPc overexpressing cells (transient or also inducible system in case of spontaneous toxicity).
- Fig1 & 2: HT1080 PrPc OE are consistently used to study MoA, use of second cell line is today standard requirement to exclude cell line specific effects. Maybe consider neuronal cell line e.g. HT22?
- Fig 3: Increased expression of FABP5, PSTG2, TFRC and HMOX1 is typically associated with ferroptosis, but not exclusive evidence. Consider 4HNE IHC staining on organoids.
- Fig4C: drop in RAC3 is observed in PRPc overexpressing cells. Wouldn't you expect then more ferroptosis resistance and drop in mesenchymal level? This is shortly discussed:
"both the organoids and the patients may be classified as "end-stage", after the majority of cell death has taken place. This would imply that cells with higher levels of RAC3 have already succumbed to prion induced cell death, and that remaining cells express lower RAC3."
Can't this be experimentally be adressed? Kinetic analysis of PrPc vs GPX8 vs Rac3?

Reviewer #3

(Remarks to the Author)

This study by Peng and colleagues explores the role of Prion protein in causing cellular toxicity. They find that overexpression of PrP in a cell line has no effect on intracellular iron levels, but leads to lower levels of oxidative stress in the ER and elevated levels and activity of antioxidant enzyme Gpx8 when exposed to hydrogen peroxide. The authors show that ferroptosis inhibitors, but not necroptosis and apoptosis inhibitors, rescue PrP-overexpression induced death. They also show that PrP overexpressing cells are more susceptible to IKE-induced ferroptosis and lipid peroxidation. Overexpression of PrP also leads to remodeling of lipid membranes and increased levels of some ferroptosis associated markers in the cell line. Brain organoids with pathogenic prion added or PrP knocked out had increased or decreased susceptibility to RSL3-induced ferroptosis, respectively. The authors perform a CRISPR screen in the cell line used previously and uncover RAC3 as a target that sensitizes the cells to prion-induced death, and show that RAC3 works synergistically with PrP to promote ferroptosis susceptibility. The authors also show that RAC3 mimics the metabolite and phospholipid alterations of IKE, and that RAC3 increases markers associated with mesenchymalization. The authors then show some immunostaining of CJD organoids and patient brains to further tie back RAC3 to disease.

This is an interesting study, further linking ferroptosis to neurodegenerative disease and uncovering a mechanism by which Prion protein causes neurotoxicity. While the study touches on these very important topics, the model systems used and conclusions drawn from them fall short of supporting the proposed mechanism. Please see specific points below:

Major points:

The cell line used for much of the paper, HT-1080 are cancer fibrosarcoma cells that are highly proliferative. Given the focus of much of the paper is related to Rac-3 and mesenchymalization, both cancer related pathways, it's unclear whether or not these findings would hold true in neurons which are postmitotic and have very different membrane dynamics. The authors should run studies to replicate these results in neuron monocultures (differentiated cell line or iPSC) exposed to and/or overexpressing Prion protein.

Alternatively, a more detailed analysis of the brain organoids would also help. Do the authors specifically see death of neurons? Is RAC3 localized to neurons in the organoids by co-staining? Do they see lipid peroxide accumulation in the organoids and in which cells? This can be achieved using fixable Bodipy stains. The authors should also use a more potent ferroptosis inhibitor (Fer-1, Lip-1, etc) to confirm that the LDH release they are seeing is caused by ferroptosis, since the aToc only partially rescued.

The hypothesis about PrP causing reduced oxidative stress in the ER leading through GPX8 upregulation, but still increasing ferroptosis susceptibility seems counterintuitive and it's unclear how it relates to the findings in the rest of the paper. The authors should try to explore how this mechanism might be working and if a connection exists. The stainings of Rac3 in the organoids and patient brains are nice, but quantification of the levels should be assessed and statistical comparisons done to draw any conclusions. The piece about potential synaptic localization because of the punctate staining pattern should be validated by co-staining with synaptic antibodies.

Reviewer #4

(Remarks to the Author)

From a macroscopic point of view the lipidomics part seems to be ok, but since details always matter in data processing (identification and quantitation by software), it would be good to get a more accurate description of this step in the workflow. The detailed description of lipid identification and quantitation could e.g. be provided in the supplement.

Version 1:

Reviewer comments:

Reviewer #2

(Remarks to the Author)

Reviewer #3

(Remarks to the Author)

The authors rely a lot on FABP5 as a marker of ferroptosis, but despite the link to their own paper, there isn't much evidence in mice or humans that FABP5 is a specific ferroptosis marker. In fact, FABP5 is widely expressed in many brain cell types and dysregulated in disease. It seems like a very useful in vitro marker, but changes in the mouse model or sCJD infected organoids don't necessarily mean ferroptosis has been induced. A lack of change in 4-HNE further suggests this might be the case, despite the authors note about its transient nature, it's one of the most stable measures of lipid peroxidation. This diminishes the connection between the stronger cell monoculture portion of the paper and the organoid work.

On a side note, in Figure 3 the Western blots are unlabeled. I assume they are in reverse order to the mRNA data, but this should be clarified. The blots should also be quantified for the replicates as individual images aren't very informative.

It's great that the authors performed analysis of protein levels in mouse models and human brain (Figures 3 and 6), but it's unclear whether the densitometry measurements are normalized to the loading control. This is important because there are differences in the bands and intensity seen for each type of sample. For example, in Figure 6b normalizing Rac3 levels to the total protein levels would likely eliminate any reduction. This is a critical point, because if the levels of Rac3 are unchanged when normalized, then the evidence in humans for this mechanism is quite weak.

Figure 6D still isn't informative because it's very unclear whether there is any overlap in the two stainings. It should just be removed from the manuscript.

Reviewer #4

(Remarks to the Author)

There are no further questions by the reviewer.

Reviewer #5

(Remarks to the Author)

Summary: The authors identify ferroptosis as a cell death pathway active in prion disease. Somewhat paradoxically, both increased PrPC expression and prion infection sensitize cells to this cell death pathway. In their response to previous reviews, the authors provided additional data that contribute significantly to the support both of these claims.

Major points:

Why is it that both the expression of PrPC and infection with PrPSc lead to an increased sensitivity to ferroptosis? As mentioned, the data supporting both claims is much stronger, but I feel that there is a critical mechanistic connection that has been overlooked. It would be of interest to determine how PrPSc is doing this but, admittedly, this is beyond the scope of the current work. However, I feel that it would be beneficial if the authors could speculate on this further, and incorporate PrPSc into the cartoon presented in Fig 6E.

Minor points:

Blots using CJD patient brains are inconsistently labeled. Are these each separate patients or from two patients as indicated in Supp Fig 8?

Line 85: "... indicating that PrPSc and RAC3 together provoke cellular loss". Do you mean accumulation of PrPSc and loss of RAC3?

Line 355: "PrPC levels have been shown to decrease over the course of prion disease" lists reference 57. This is the incorrect reference for this claim. The correct reference: Mays CE, Kim C, Haldiman T, van der Merwe J, Lau A, Yang J, Grams J, Di Bari MA, Nonno R, Telling GC, Kong Q. Prion disease tempo determined by host-dependent substrate reduction. The Journal of clinical investigation. 2014 Jan 16;124(2).

I would suggest a careful checking of references.

Version 2:

Reviewer comments:

Reviewer #3

(Remarks to the Author)

I don't have any further comments for the authors and thank them for spending time to perform the additional requested experiments.

Reviewer #5

(Remarks to the Author)

I have no further comments regarding the experiments presented in this manuscript.

Some additional editorial points:

Abstract: "Consequently, native as well as infectious CJD prions triggered ferroptotic markers and sensitization." The term "prion" is reserved for the disease causing agent formed by the prion protein. For clarity, I would change this sentence to "Consequently, both PrPC and infectious CJD prions triggered ferroptotic markers and sensitization." There is also mention of "native prions" on line 224.

Introduction "Scrapie in animals" should read "scrapie in sheep"

Thanks to all the reviewers for their helpful comments and suggestions. We believe they strengthened the manuscript substantially.

Reviewer #1 (Remarks to the Author):

Summary: The authors investigate a cell death pathway previously unattributed to prion disease, ferroptosis. They show that PrPC sensitizes cells to ferroptosis by maintaining cells in a low redox state, which allows for the accumulation of unsaturated long-chain phospholipids. These lipids are susceptible to peroxidation during bursts of oxidative damage, which ultimately leads to ferroptosis. The authors link the expression of PrPC with that of GPX8 and RAC3, which exacerbate this sensitivity. They present correlates of this in CJD infected cerebral organoids, and in the brains of CJD patients.

Major points:

This paper is exceedingly difficult to read, and its overall message is obscure. There is no attempt to connect the findings to a simple, logical model or hypothesis. To suggest that PrPC and PrPSc both drive susceptibility to the same cell death pathway (ferroptosis) is difficult to accept, and comes with a high burden of proof, which is not met. The data supporting a connection between PrPSc and ferroptosis is particularly weak.

Apologies if the manuscript was unclear. We agree that the concept and terminology can be dense in places and we attempted to streamline these into a more coherent message. Redox biology is complex, and many conclusions are multifaceted and cannot be communicated in a simple soundbite. Thus, we appreciate the extra burden on reviewers to elucidate these key points for the scientific community. We also hope that the schematic in Fig 6 clearly conveys the overall message.

PrPSc drives ferroptosis susceptibility:

1. p6 line 146: regarding CJD infected organoids: "... subsequently die of ferroptosis." This statement is not well justified. The data show only an increased staining of FABP5. Can this be quantified? How many organoids were investigated? Staining of CJD patient samples (Figure 6) would be helpful here. Thanks for the suggestion, FABP5 is indeed difficult to quantify in a snapshot and we are happy to further support the conclusion. Depending on the timing, cohorts of cells may be expressing FABP5 in at different levels during ferroptosis. At later timepoints, levels may be lower due to already-dead and missing cells. It's also unlikely that all organoids undergo death at the same rate, due to size and consistency.

The reviewer is therefore correct and human CJD samples are preferable. We took available normal brain homogenates and CJD tissue to examine FABP5 and found a pronounced increase in all instances (see new Fig. 3E)

We moreover infected mice with scrapie (RML model) to analyze a timecourse of the disease and saw a consistent and distinct increase in FABP5 protein in brain homogenates over time (new Fig. 3D).

a. Fig 3D: was the increased sensitivity to RSL-3 induced LDH release, and rescue by aToc, statically significant?

We included the missing statistics for this figure subpart and corrected the RSL-3 concentrations which mistakenly had an extra '0'.

b. Fig 3E shows an increase in LDH release upon RSL-3 treatment of CJD infected organoids compared to non-infected organoids. This is all taken to support the conclusion that “infectious prions drive ferroptosis susceptibility”. However, an alternative explanation is that these organoids are more susceptible to RSL-3 because they are already under the stress of prion infection, as opposed to a direct result of PrPSc itself. A demonstration of their response to stimuli promoting other forms of cell death would strengthen this conclusion.

Thanks for this valid point. We treated organoids with staurosporine, a classic inducer of apoptosis and other nonspecific forms of cell death. We used an LDH assay at 72 hr to allow for sufficient cell death with normal brain homogenates and CJD treated organoids but observed no significant differences between NBH- and CJD-treated organoids. We included these data in Supplementary Figure 4C. Please see also Supp. Fig. 2F.

2. Figure 6: The reduction of RAC3 staining in the frontal cortex of CJD patients is the only evidence tying these findings to CJD patients, and prions in general, despite this being in the title of the manuscript. Comparing the staining intensity of brain slices is inherently subjective – can a western blot be performed? It would be useful to analyze other proteins, as mentioned above, such as FABP5 and GPX8 to help substantiate these findings.

Thanks for the suggestion. Clinical case sections are processed by the pathological laboratory at Goethe University Frankfurt on a fully automated Leica Bond III using the Leica Bond Polymer Refine Detection Kits, which has been shown to be very consistent between slides. Nevertheless, as the reviewer points out, staining intensity in slices is difficult to quantify, thus we refer to our reply to Q1, in which FABP5 in CJD patient brains is additionally shown by Western. We used the same material to quantify GPX8 and also observed a marked albeit statistically insignificant increase. We included these data are now in Supp. Fig. 8A.

a. How many organoids were used to look at RAC3? Can this be quantified? I find these data (Sup Fig 7A) unconvincing in their present form.

Yes, we analyzed it with the other mentioned markers in the next question.

3. The reduction of the levels of a protein in the brain of an individual who died of a neurodegenerative disease can be explained in many ways. A time series looking at the levels of RAC3, FABP5, GPX3 and PrP^{Sc} over the course of disease (organoids, mice, or brain slices) would help demonstrate the specificity of these changes to the development of prion disease.

Great suggestion, thanks for the insight. For obvious reasons, timecourses are not possible with human material. We again chose to use PrP^{Sc} infected mouse brains and observed the same distinctive loss of RAC3 protein, please see new Fig. 6A. We saw the same loss of RAC3 protein in

human CJD cases, supporting conservation of this mechanism (new Fig. 6B). We added a few more images of RAC3 stained organoids to the Supplement 8A. For FABP5 and GPX8 we refer to the above answers.

PrPC drives ferroptosis susceptibility:

The data connecting PrPC to ferroptosis is somewhat more convincing than that for PrPSc, but the experiments are done in a cell line that is highly susceptible to ferroptosis. Can these observations be recapitulated in other cell lines, or (perhaps more importantly) in neurons? One would expect from these data that a comparison of Prnp knockout neurons vs WT neurons vs PrP overexpressing neurons would show increasing susceptibility to ferroptosis inducing factors like RSL-3 or erastin. This is an important question raised by reviewers and one that we wanted to examine ourselves in more detail. The subject of ferroptosis in neuronal cell death has been investigated by many labs in the last few years. Neurons in particular have a very high component of polyunsaturated fatty acid containing phospholipids in the membrane which must be protected with an antioxidant system. Aging brains also frequently have an imbalance of iron, which can drive sensitivity and many different cell types. Due to the extensive research on the subject, we referred to several recent reviews that synthesize the primary literature supporting ferroptosis in neurons (PMIDs: 33536413, 38395733, PMC9170779, 37252357, 38367511, PMC11176077).

One of the key cell death mechanisms well known in the nervous system is glutamate toxicity. Under oxidative conditions, cystine uptake via the System Xc⁻ is accompanied by the release of glutamate. Cells experiencing imbalances in oxidative stress frequently take up more cystine because it is a building block for glutathione, which is essential for the lipid peroxidation detoxification via GPX4. However, System Xc⁻ is an antiporter, thus each molecule of cystine taken up will export one molecule of glutamate. These locally elevated glutamate levels can lead to excitotoxicity in various neurodegenerative diseases including ALS, Alzheimer's and Huntington's disease. There are multiple

routes to the glutamatergic excitatory input on neurons, ultimately resulting in ferroptosis. Notably, glutamate toxicity is blocked with ferroptosis inhibitors, although this may be a circuitous mechanism due to decreased requirements for glutathione.

In our hands the presence of serum is a prerequisite for ferroptosis, thus precluding most protocols for iPS propagation and differentiation of neuronal cell lines such as SY5Y. We previously attempted to generate neurons from such lines and then switch them to FBS-containing medium but they suffer a serum shock and are only partially rescued by ferroptosis inhibitors.

For this reason, we chose to address the question below as suggested by Rev#2 using HT-22 cells grown in 10% FBS as a model. Please see as well also the answer to question # 5 below showing colocalization of FABP5 and MAP2 in neuronal subsets of organoids.

1. Figure 1A: Iron balance is unaffected by the expression of PrPC. This is a surprising result, considering that the rest of the data connects PrPC expression to ferroptosis. This should be discussed in detail. Is this because of the reduction in PrP expression over time? If so, this analysis bears repeating with the DOX inducible cells.

Thanks for the insightful comment. Classically, ferroptosis is dependent on the stoichiometry of iron, polyunsaturated fatty acid containing phospholipids, and antioxidants. Tipping the balance of one of them does not require that the others must be perturbed. Since PUFA phospholipids are increased, this would be the chink in the armor, thus iron levels can remain constant and still catalyze the Fenton reaction. It is not due to expression reduction over time as we use freshly infected cells, validated by Western and immunostaining, for these analyses. Please also see the response to Rev#2 below for a larger discussion of how to accurately measure iron in cells and new results.

2. The connection between PrP expression and a reduction in cell viability is somewhat casually introduced, with supporting data not displayed until later in the manuscript (Fig 2A). This would benefit from some restructuring. Further, while the use of resazurin shows a reduction in the viability of PrP overexpressing cells, a live/dead assay would better support the statement that these cells are “lost” over time.

Thanks for the suggestions. The reviewer is correct in that ordinarily the phenotype is shown first, followed by the mechanism. In this manuscript we first observed that stable iron concentrations are found in PrPc expressing cells, in contrast to in particular the results of one group. Nevertheless, the balance of oxidative stress has changed, leading to the logic that ferroptosis should be tested. We decided to smooth concepts from Fig1/2 together by including additional data on GPX8 in PrP^C-expressing cells in Fig. 2, referred to in the next question.

Resazurin detects metabolism in live cells, so a live/dead cell assay is important. We performed this test by flow cytometry using Calcein-AM, a cytosolic dye that is lost upon membrane permeabilization, and propidium iodide, which stains nuclei in compromised cells. A clear loss of viability from RSL-3 treatment as indicated by both markers is observed in the Q3 quadrant of the dot plots, while Fer-1 protects this population (new Supp. Fig. 2D).

3. Figure 1 G-H: The concomitant increase in GPX8 expression along with PrPC, and subsequently increased peroxidase activity and lower H₂O₂ levels could be explored further. For example, does GPX8 knockdown in PrPC overexpressing cells return the peroxidase activity and H₂O₂ levels to those observed in non-PrPC overexpressing cells?

This is an interesting point also raised by the other reviewers. We attempted this experiment by knocking down GPX8 in PrPc overexpressing cells, but cell viability suffered, likely due to increased oxidative stress. We believe any surviving cells would therefore show a bias on any outcome on H₂O₂. Therefore, we chose to test GPX8 knockdown in control cells together with Hyper3-ER H₂O₂ sensor. The results were consistent with the overexpression experiments; KD revealed an increase in luminal H₂O₂ (new Supp. Fig. 2A).

We moreover included GPX8 knockout (which survive up to 3d) in the background of PrPc (doxycycline-treated) cells and observed an increase in ferroptosis susceptibility 1d after viral expression (also evinced by loss of GPX8 protein). We included these data now in Fig. 2B.

4. Figure 2C: Why don't PrPC overexpressing cells demonstrate higher levels of lipid peroxidation without RSL3 induction? Shouldn't this be the case, given that they have increased staining for FABP5 (Figure 3A)?

This is a fascinating point. While it may seem initially counterintuitive, it makes sense in the constellation of redox signaling. Simply put, we believe that cells with lower levels of ROS are more susceptible to ferroptosis due to increased PUFA-PLs in the membrane, while surviving cells with higher ROS levels have established an antioxidant system to cope with those levels.

As shown in Fig. 2A, Fer-1 protects PrP-^{OE} cells, but the window in which to see this extends over several days, while lipid peroxidation is a very quick and terminal reaction. Thus cellular basal levels do not show an increase in PUFA peroxidation. However, FABP5 is an early marker of ferroptosis and may show potential susceptibility by increased PUFA-PLs (PMID: 38653992). We currently have 2 separate manuscripts in preparation that show this phenomenon in more detail for other genetic manipulations.

We further suspect that cells having a high complement of lipids favoring ferroptosis do not require high levels of lipid peroxidation in order to undergo ferroptosis, and they may actually require much lower and produce low, if any, Bodipy signal. What we didn't show is that a larger fraction of PrP^C-^{OE} cells have died at this four-hour RSL3-treatment timepoint. They died evidently because a small burst of lipid peroxides was enough to kill them extremely rapidly. We did 1 hr timecourses but the ratio was the same. We show the whole FACS plots with this decrease in viable cells in Supp. Fig. 2E.

a. Figure 3A: A western blot for FABP5 would be more convincing. It would also be nice to see confirmation of the qPCR data by western blotting for PSTG2, TFRC, and HMOX1.

Thanks for the comment, we agree it is important to control both mRNA and protein levels.

We performed Westerns on the same cells and observed substantial increases in all cases except for TFRC. This may imply a form of auto-regulation or compensation by these cells, which further supports unchanged levels of iron in these cells. We also included a Western for lipid degradation product 4-Hydroxynonenal (4-HNE) which shows a striking increase likely due to tonic ferroptosis in PrP^C OE cells. We include these data in Fig. 3B.

5. The data presented in Sup.4B of single organoids (WT vs PRNP KO) treated with the ferroptosis inducer RSL3 are not convincing. Are there additional images, western blots, or quantification that can bolster the conclusion that there is reduced FABP5 staining in PRNP KO organoids?

Thanks for the suggestion. To support this argument for FABP5 we used *in vivo* data as described for the questions above in mouse and human (Fig. 3 new panels). We removed the poor-quality images in Sup. Fig. 4B. We additionally prepared a new panel of stainings to demarcate FABP5 expression in organoids. We found that the highest concentrations in WT organoids are coincident with MAP2 (neuronal) expression rather than GFAP (glia), suggesting it is these regions which are most susceptible to ferroptosis. We include these data as a new Supp. Fig. 5.

6. Figure 4C: Are these differences statistically significant? The PrP signal in RAC3 overexpressing cells does not match the quantification. On this note, the western blots for PrP show only a single band – PrP typically migrates as several bands due to differential glycosylation and physiological cleavage events.

Thanks for the comment. Actually the PrP^C actin band in OE is considerably more intense than the control band, resulting in less PrP^C overall. The qPCR confirms this now with statistics. As a note, we observed PrP^C migrating as 2 separate bands in the Western, consistent with the di-glycosylated species being the dominant form, with some mono-glycosylated that may be still transiting the secretory pathway.

Minor points:

p2 line 23: “Although the precise mechanism by which the prion protein kills neurons...” This should read: “... by which prions kill...” the prion protein (PrP^C) is not in and of itself toxic to neurons.

Thanks for this clarification

p3 line 66: “We observe that cells constitutively expressing PrP^C lose expression over time, despite selection with a dicistronic puromycin resistance gene, while PRNP (the gene encoding PrP^C) knockout cells grow faster (Fig. 1B, Supp. Fig. 1C)”. It is not clear how these statements are related to one another through experimentation.

PrP^C expression is clearly lost as measured in Fig. 1B. The experiment demonstrating growth was established by the Chronos Gene Effect, which is in essence CRISPR KO screens in >1000 cell lines. This can be visualized in the DepMap database, where each data point indicates a cell line. The guide distribution of *PRNP* guides is then benchmarked against all other guides, and their relative frequency is extrapolated versus the entire library population. Genes having “Gene Effect” values greater than zero indicate better growth/survival, while values less than zero indicates loss of viability or growth. Practically all human genes are included in the library, thus relative effects on growth can be easily deciphered. TP53 and beta-actin are included as controls in Supp. Fig. 1 with their relative effects on growth. Since our work primarily involved in the cell line HT1080, we included that point in the excerpt below, also showing improved growth/survival.

p4 line 76: “As an endoplasmic reticulum resident protein, GPX8 detoxifies H₂O₂ generated during disulfide isomerase mediated protein (re-)folding and is tightly associated with mesenchymalization processes. Critically, the prion protein has shown a susceptibility to misfolding.” This statement is unnecessary and implies that GPX8 is somehow involved in PrP misfolding, which, as far as I am aware, is not the case.

Thanks for the point. Indeed this has not been shown, we have corrected the statement.

p5 line 134: The reader is directed to Supp Fig3A regarding FABP5 expression, but this figure shows RT-QulC assays of CJD samples.

Corrected

p6 line 146: "...subsequently die of ferroptosis.". It has been established that these organoids show higher levels of FABP5, but not that they subsequently died of ferroptosis.

Corrected

p7 line 174: " A highly depleted hit, glycosyltransferase ABO, was recently implicated in Covid pathogenesis via its glycosylation activity, a process that is known to affect prion localization and function." The reference (44) shows only that PrP is glycosylated, not a connection with this particular glycosyltransferase, or SARS-CoV-2 pathogenesis.

We removed this implied linkage

The role of PrP in EMT has been described previously (Mehrabian, et al. "The prion protein controls polysialylation of neural cell adhesion molecule 1 during cellular morphogenesis." PLoS One 10.8 (2015): e0133741). This paper should be discussed.

Thanks, we missed this reference and have now included it in the discussion.

Reviewer #2 (Remarks to the Author):

Thank you for the insightful and helpful review.

Peng et al report on novel MoA of how prion induces degeneration through ferroptosis via reducing ER-ROS involving GPX8 and RAC3 driven mesenchymalization.

- Intro or discussion. Please include recent pioneering report linking YBX1/RAC3 axis to ferroptosis (10.1038/s41419-024-06882-5)

Certainly we included this paper in the discussion.

- Fig. 1: Please include positive and negative ctrl also using FerroOrange. An CE-ICP-MS approach measuring ratios Fe^{3+}/Fe^{2+} would be ultimate proof and recommended.

Thank you for this insightful comment. We agree that FerroOrange is an indirect method and that specific measurement of iron is essential. We included the control experiment in Supp. Fig. 1A and performed a direct measurement as suggested.

Most existing methods for measuring the Fe^{3+}/Fe^{2+} ratio are based on disruptive techniques in cell lysates. During cell lysis, there are unpredictable transformations of Fe^{2+} and Fe^{3+} , which can lead to inaccurate determinations of this ratio. Numerous reducing agents present in cells can reduce Fe^{3+} to Fe^{2+} , especially when Fe^{3+} is released from proteins (e.g., ferritin); these transformations are highly dependent on pH. Conversely, Fe^{2+} is very sensitive to oxidation by atmospheric oxygen (PMID 30931301).

All such transformations are under complex thermodynamic and kinetic control, making them difficult to regulate due to numerous influencing factors, such as resident compounds, temperature,

and pH. Mössbauer spectroscopy does not have these drawbacks, as it is a non-disruptive method (10.1515/nuka-2017-0024). Additionally, the use of liquid nitrogen ($T = 85 \text{ K}$) helps to quench potential transformations, mostly due to kinetic restriction of such reactions. In a personal communication from Bernhard Michalke, who developed the CE-ICP-MS method for measuring the $\text{Fe}^{3+}/\text{Fe}^{2+}$ ratio (PMID 30931301), he claims that this method is extraordinarily sensitive and most laboratories cannot reproduce their own results accurately. CE-ICP-MS method is disruptive because the measurements are conducted in cell lysates and the results are highly sensitive to the pH and presence of oxygen. Therefore we chose to use Mössbauer spectroscopy for its accuracy and consistency.

Our measurements using Mössbauer spectroscopy have shown that the Fe^{3+} concentration was not changed in PrPc OE samples. Other papers show about 100 times bias in labile iron pool (LIP) measurements by disruptive methods: "Methodological problems associated with disruptive methods are reflected in the wide range of LIP concentrations (3.5–230 μM) obtained with these methods for the same types of cells and tissues..." (PMID 14637247).

"The major disadvantage of these approaches is decompartmentalisation of cellular iron and change in the $\text{Fe}^{2+}/\text{Fe}^{3+}$ ratio due to oxidation of iron during fractionation, disruption of iron-containing proteins and/or release of iron from iron storage proteins" (PMID 14637247). This spurious detection of Fe^{3+} to Fe^{2+} "may be that the high level of Fe^{2+} detected by spectrophotometry... is caused by homogenization of the sample ... in the presence of hydrochloric acid and pepsin, which might have destroyed the protein shell of ferritin and released iron from the ferritin iron core." (PMID 8771061).

In the original paper "Prion Protein Regulates Iron Transport by Functioning as a Ferrireductase" (PMID 23478311), the authors point out that the ferrireductase "activity in lysates was 9-fold higher than cell surface in both cell lines probably due to contribution from other intracellular FR proteins". The authors demonstrated that ferrireductase activity in cell lysates increased ninefold in both PrPc overexpression and normal cells. It is possible that the authors observed an indirect effect of PrPc OE on the reduction of Fe^{3+} in cells damaged by the analytical conditions, which resulted in the decompartmentalization of reducing agents that are elevated in PrPc OE.

The following shows the data from our measurements which are now included in a new Supplemental File 1.

Measurements of ^{57}Fe enriched samples

Mössbauer spectroscopy on 3 replicates of both HT-1080 control and PrP^C OE samples (grown on ^{57}Fe enriched $^{57}\text{Fe}^{3+}$ -citrate solution for 1 week with media changes each 2 days) were measured to characterize the iron compounds present in the samples.

^{57}Fe Mössbauer spectroscopy measurements were performed at liquid nitrogen temperature ($T = 85 \text{ K}$) in a bath cryostat, using a conventional Mössbauer spectrometer (WissEl, Starnberg, Germany) operating in constant acceleration mode with ^{57}Co source in Rh matrix.

Spectra were evaluated with the assumption of Lorentzian line shape by standard computer-based statistical analysis methods that included fitting the experimental data using a least-squares minimization procedure with the help of the MOSSWINN program (Klencsár et al. 1996). The parameters calculated for the spectral components correspond to hyperfine parameters of Mössbauer nuclei such as isomer shift (δ), quadrupole splitting (Δ), linewidth (Γ) and partial resonant absorption areas (Sr%). ^{57}Fe isomer shifts are given relative to α -iron at room temperature. Mössbauer spectroscopy is a non-destructive method to determine the electronic structure (oxidation and spin state) and binding sites of iron, thus, the $\text{Fe}^{2+}/\text{Fe}^{3+}$ ratio.

The characteristic Mössbauer parameters (at $T=85\text{K}$) of high spin Fe^{3+} coordinated by O-donor ligands typically fall into the range of $\delta=0.4\text{-}0.5 \text{ mm/s}$ and $\Delta=0.4\text{-}0.8 \text{ mm/s}$ while those of high spin Fe^{2+} (in similar environment) are of $\delta=1.1\text{-}1.4 \text{ mm/s}$ and $\Delta=2.0\text{-}3.4 \text{ mm/s}$ (Vértés et al. 1979; Greenwood and Gibb, 1971).

Mössbauer spectroscopy was successfully applied in the case of biological samples e.g. in the study of blood cells, brain tissues, etc. (Galazka-Friedman et al., 1996, 2008, 2009; Bauminger et al. 1987; Rzepecka et al., 2017). According to these papers (and further references therein), typical Mössbauer parameters of Fe^{2+} -hemoglobin are $\delta=0.2\text{-}0.3$ mm/s and $\Delta=2.1\text{-}2.3$ mm/s while Fe^{3+} found in ferritin are $\delta\sim 0.5$ mm/s and $\Delta=0.6\text{-}0.7$ mm/s.

Mössbauer spectra results:

Mössbauer spectra obtained at $T=85\text{K}$ of control and PrPC OE samples. Dots represent the experimental data; solid lines are the computer fits to the experimental data. Black line above the spectrum shows the residual of the experimental and fitted data.

Control samples

δ (mm s⁻¹)
 Δ (mm s⁻¹)
 Γ (mm s⁻¹)
 Sr(%)

Doublet₁

0.468(7)
 0.73(1)
 0.49(2)
 100

PrP samples

δ (mm s⁻¹)
 Δ (mm s⁻¹)
 Γ (mm s⁻¹)
 Sr(%)

Doublet₁

0.468(7)
 0.73(1)
 0.53(2)
 100

Mössbauer parameters obtained from the fitting of the spectra of control and PrPC OE samples. Errors (in the last digits) are given in parentheses. δ : isomer shift; Δ : quadrupole splitting; Γ : full linewidth at half maximum; Sr: partial resonant absorption areas of spectral components which represent relative contents of the corresponding iron form.

According to the Mössbauer parameters, the Doublet₁ can represent a high spin Fe^{3+} -compound, in distorted octahedral O₆ coordination and it can be probably assigned to ferritin (and/or hemosiderin) (Galazka-Friedman et al. 1996, Vértes et al. 1979). The presence of high spin and low spin Fe^{2+} (e.g. in hemoglobins or Fe^{2+} -complexes) cannot be found since they have completely different parameters. The spectrum of the control and PrP^C OE samples are identical.

However, since the absorption effect is rather low (due to the very low ⁵⁷Fe quantity), approx. up to 10% Fe^{2+} as minor compound is not detectable under these conditions. (If we simulate 10% Fe^{2+} with average parameters e.g. $\delta=1,3$ mm/s and $\Delta=2,9$ mm/s in the case of any of the spectra; the normalized chisquare value of the fitting will slightly increase from 1.044 to 1.080 (PrP^C OE samples).

Spectrum of PrPC OE samples with suggested high spin Fe²⁺ compound (10%). The blue line represents the original Fe³⁺, while green line indicates Fe²⁺.

However, one must point out that the result of this assumption depends much on the chosen parameters that were fixed in the simulation, thus it might be rather a raw estimation. Measurements using Mössbauer spectroscopy have shown that iron present in the same state (mostly Fe³⁺, such as ferritin (and/or hemosiderin)) in the control and PrP^C OE. The Fe³⁺ concentration was not changed in PrP^C OE samples compare to control. This confirms there was not observed any significant conversion of Fe³⁺ to Fe²⁺ in PrP^C OE cells.

• Fig. 1: To strengthen conclusion on involvement of GPX8, one should perform knockdown of GPX8 in PrPc overexpressing cells (transient or also inducible system in case of spontaneous toxicity). Yes we were very happy to do this, however GPX8 knockouts are nonviable after a few days so we could only measure in a snapshot shortly following infection of a Cas9/GPX8 guide-containing virus. We nonetheless could observe increased sensitivity in GPX8 KO to ferroptosis in the presence of PrP^C (shown in new data Fig. 2B), further supporting their synergy. Thanks for the comment.

• Fig1 & 2: HT1080 PrPc OE are consistently used to study MoA, use of second cell line is today standard requirement to exclude cell line specific effects. Maybe consider neuronal cell line e.g. HT22?

This is a highly valid observation pointed out as well by the other reviewers. We overexpressed PrP^C and knocked out *Prnp* in mouse HT22 and observed the same effect as in HT-1080 cells, supporting a

shared mechanism between mouse and humans further detailed in new data in Fig. 3. We were happy to include these data in Supp. Fig. 6A.

Moreover, although the PrP^C and RAC3 OE results were very clear in HT1080 and HT22 cells, we didn't formally demonstrate RAC induced mesenchymalization in humans brains or neurons. Therefore we removed the "driven mesenchymalization" portion of the title.

• Fig 3: Increased expression of FABP5, PSTG2, TFRC and HMOX1 in typically associated with ferroptosis, but not exclusive evidence. Consider 4HNE IHC staining on organoids.
 Thanks for the comment. We have supplemented the data with Westerns of these molecules in new Fig. 3 and performed 4HNE stains on the organoids as requested. However, 4HNE as a chemical byproduct is somewhat transient and it is likely that the snapshot does not capture the full range of 4HNE production through lipid degradation products. For example, does less 4HNE mean less lipid peroxidation overall, or does it mean that the cells have already disappeared? We tested in several organoids but the results proved inconclusive to warrant further investigations, although a mild yet nonsignificant increase is observed. We included these stainings in Sup Fig 4B. Please also see the reply to Rev#1, Q4a, above.

• Fig4C: drop in RAC3 is observed in PRPc overexpressing cells. Wouldn't you expect then more ferroptosis resistance and drop in mysenchymal level? This is shortly discussed:
 "both the organoids and the patients may be classified as "end-stage", after the majority of cell

death has taken place. This would imply that cells with higher levels of RAC3 have already succumbed to prion induced cell death, and that remaining cells express lower RAC3.”

Can't this be experimentally be adressed? Kinetic analysis of PrPc vs GPX8 vs Rac3?

An insightful and similar point to Rev#1 Q3. We chose to analyze this more conclusively in timecourses of infected animal brains, please see the reply above.

Reviewer #3 (Remarks to the Author):

This study by Peng and colleagues explores the role of Prion protein in causing cellular toxicity. They find that overexpression of PrP in a cell line has no effect on intracellular iron levels, but leads to lower levels of oxidative stress in the ER and elevated levels and activity of antioxidant enzyme Gpx8 when exposed to hydrogen peroxide. The authors show that ferroptosis inhibitors, but not necroptosis and apoptosis inhibitors, rescue PrP-overexpression induced death. They also show that PrP overexpressing cells are more susceptible to IKE-induced ferroptosis and lipid peroxidation. Overexpression of PrP also leads to remodeling of lipid membranes and increased levels of some ferroptosis associated markers in the cell line. Brain organoids with pathogenic prion added or PrP knocked out had increased or decreased susceptibility to RSL3-induced ferroptosis, respectively. The authors perform a CRISPR screen in the cell line used previously and uncover RAC3 as a target that sensitizes the cells to prion-induced death, and show that RAC3 works synergistically with PrP to promote ferroptosis susceptibility. The authors also show that RAC3 mimics the metabolite and phospholipid alterations of IKE, and that RAC3 increases markers associated with mesenchymalization. The authors then show some immunostaining of CJD organoids and patient brains to further tie back RAC3 to disease.

This is an interesting study, further linking ferroptosis to neurodegenerative disease and uncovering a mechanism by which Prion protein causes neurotoxicity. While the study touches on these very important topics, the model systems used and conclusions drawn from them fall short of supporting the proposed mechanism. Please see specific points below:

Many thanks for the thorough and fair review. Please note that for brevity's sake in several instances we refer to related answers given to the other reviewers above.

Major points:

The cell line used for much of the paper, HT-1080 are cancer fibrosarcoma cells that are highly proliferative. Given the focus of much of the paper is related to Rac-3 and mesenchymalization, both cancer related pathways, it's unclear whether or not these findings would hold true in neurons which are postmitotic and have very different membrane dynamics. The authors should run studies to replicate these results in neuron monocultures (differentiated cell line or iPSC) exposed to and/or overexpressing Prion protein.

Indeed, a great question also asked by the other reviewers, please see the replies to Rev#1&2 queries and suggestion of HT-22. We moreover examined the localization of FABP5 positivity in organoids in the new Supp. Fig. 5 and found a high degree of overlap with MAP2 positive regions compared to GFAP, suggesting it is these regions that succumb to cell death.

Supplementary Figure 5

Alternatively, a more detailed analysis of the brain organoids would also help. Do the authors specifically see death of neurons? Is RAC3 localized to neurons in the organoids by co-staining? Do they see lipid peroxide accumulation in the organoids and in which cells? This can be achieved using fixable Bodipy stains. The authors should also use a more potent ferroptosis inhibitor (Fer-1, Lip-1, etc) to confirm that the LDH release they are seeing is caused by ferroptosis, since the aToc only partially rescued.

Yes, a good insight, also addressed above with 4HNE staining mentioned above. However, we have the problem of the transient emergence of 4HNE following lipid peroxidation. We tried fixable Bodipy stains but the result was not improved. We also tried other inhibitors (Lip-1, and Fer-1 which worked well, e.g., Fig 2A) in the monolayer cultures but we believe we have a diffusion problem due to the thickness of organoid, or alternatively, partial ferroptosis in combination with another form of cell death. This is supported by the eventual development of hypoxic cores indicating susceptible inner cells are deprived of oxygen and nutrients that may synergize with RSL-3. We cannot however rule out the presence of an alternative form of cell death as other inhibitors also would have a diffusion problem.

Regarding RAC3, we performed fluorescent co-staining staining together with Synaptophysin, a membrane protein in synaptic vesicles of neurons. However, using various tissues and antibodies, immune-staining was unsuccessful, even after tyramide signal amplification. Therefore, we opted for light microscopic staining for RAC3 and Synaptophysin on serial sections of adult human brain following established and validated protocols in the neuropathology department of Frankfurt University. The resulting staining was assessed by a consultant for neuropathology. While falling short of formal proof of an overlap of both markers, we established that the staining patterns of neuropil structures appear highly similar for RAC3 and Synaptophysin, indicating co-localization.

Synaptophysin

RAC3

The hypothesis about PrP causing reduced oxidative stress in the ER leading through GPX8 upregulation, but still increasing ferroptosis susceptibility seems counterintuitive and it's unclear how it relates to the findings in the rest of the paper. The authors should try to explore how this mechanism might be working and if a connection exists.

The stainings of Rac3 in the organoids and patient brains are nice, but quantification of the levels should be assessed and statistical comparisons done to draw any conclusions. The piece about potential synaptic localization because of the punctate staining pattern should be validated by co-staining with synaptic antibodies.

Please see the reply above regarding oxidative stress Rev#1 Q4. Regarding the mechanism we were also curious about this because we had noticed that GPX8 down regulates surface PrP^C (new Supp. Fig. 2B). We took your advice and investigated further: evidently GPX8 facilitates degradation of PrP^C, because it is lost when they are co-transfected. Interestingly, this effect can be reversed by addition of MG132, a proteasome inhibitor. Our interpretation of this is that a favorable redox environment allows PrP^C to be successfully degraded (following re/folding?), which favors cellular health. This is certainly worth exploring in a new study to determine whether this is also the case for infectious PrP^{Sc}. We included these new data in Supp. Fig. 2C.

Regarding Rac3: An important question, please see the response also to Rev#1 above. We believe it is important to distinguish between chronic oxidative stress and a pulse of oxidative stress. Sensitized cells in culture, due to metabolic imbalance of lipids, iron, or antioxidants, may only survive if they have lower ROS levels. Yet they may be exquisitely sensitive to ROS as evidenced by long-term protection with Fer-1 (Fig. 2). In contrast, cells with tonic high basal levels of ROS have adapted the antioxidant system or lipids to cope with the stress. Thus the former situation is more likely to respond to a strong pulse of ROS like ferroptosis induction, or possibly infection, trauma, etc. This is exactly how we envisage prion-misexpressing cells to respond. We clarified this further in the 2nd paragraph of the discussion.

With regard to the 2nd question, we quantified RAC3 in Western blots from CJD patient brains and the localization question we addressed as well in the answer above.

Reviewer #4 (Remarks to the Author):

From a macroscopic point of view the lipidomics part seems to be ok, but since details always matter in data processing (identification and quantitation by software), it would be good to get a more accurate description of this step in the workflow. The detailed description of lipid identification and quantitation could e.g. be provided in the supplement.

Thanks for the review and suggestion, we expanded the section to include further details. Because we use different machines a slightly modified protocol was applied for the two different lipidomics experiments. These are further detailed in the new Supp. Fig. 9 as well as in the Methods.

Lipid annotation and quantification workflow

Thanks to the reviewers for their helpful comments and suggestions. We believe they strengthened the manuscript substantially.

Reviewer #3 (Remarks to the Author):

The authors rely a lot on FABP5 as a marker of ferroptosis, but despite the link to their own paper, there isn't much evidence in mice or humans that FABP5 is a specific ferroptosis marker. In fact, FABP5 is widely expressed in many brain cell types and dysregulated in disease. It seems like a very useful in vitro marker, but changes in the mouse model or sCJD infected organoids don't necessarily mean ferroptosis has been induced. A lack of change in 4-HNE further suggests this might be the case, despite the authors note about its transient nature, it's one of the most stable measures of lipid peroxidation. This diminishes the connection between the stronger cell monoculture portion of the paper and the organoid work.

Thanks for the comments. Indeed, the specificity of FABP5 for ferroptosis has not been evaluated in diverse pathologies. Lipid peroxidation was tested extensively in mouse prion models using malondialdehyde (MDA) and 4-hydroxyalkenals (HAE) (Brazier et al., 2006, FRBM; manuscript lines 72-74 of the revised manuscript), which show peroxidation came up early and dropping after significant damage was done. Thus, achieving the correct time points is challenging. In accordance with this study, we examined much earlier organoids and were surprised to find a significant increase at 56 dpi. We included these data now in Supplemental Figure 4B. It is possible that lipid peroxidation products at later time points are lost due to degradation of protein adducts.

On a side note, in Figure 3 the Western blots are unlabeled. I assume they are in reverse order to the mRNA data, but this should be clarified. The blots should also be quantified for the replicates as individual images aren't very informative.

Thanks for the comment. We corrected this oversight and included quantification as requested.

It's great that the authors performed analysis of protein levels in mouse models and human brain (Figures 3 and 6), but it's unclear whether the densitometry measurements are normalized to the loading control. This is important because there are differences in the bands and intensity seen for each type of sample. For example, in Figure 6b normalizing Rac3 levels to the total protein levels would likely eliminate any reduction. This is a critical point, because if the levels of Rac3 are unchanged when normalized, then the evidence in humans for this mechanism is quite weak.

Thanks for the insight. The band densitometry measurements are normalized to the total protein densitometry – this information is on line 463 of the revised manuscript “ImageJ 1.52n was used to measure the densitometry density of the bands as well as the total protein. The levels of test protein were normalized to the total protein, and the average levels between CJD and control groups were compared by Welch’s unpaired t test (Two-tailed with 95% confidence interval).” We agree that the banding does look different; this is likely due to the degeneration associated with advanced prion disease that the control brains (people who died of non-neurodegenerative causes) do not have.

Figure 6D still isn't informative because it's very unclear whether there is any overlap in the two stainings. It should just be removed from the manuscript.

Yes, we struggled technically with co-immuno stainings and are glad to remove this unsatisfactory figure portion.

Reviewer #4 (Remarks to the Author):

There are no further questions by the reviewer.

Thank you for your comments and assistance.

Reviewer #5 (Remarks to the Author):

Summary: The authors identify ferroptosis as a cell death pathway active in prion disease. Somewhat paradoxically, both increased PrPc expression and prion infection sensitize cells to this cell death pathway. In their response to previous reviews, the authors provided additional data that contribute significantly to the support both of these claims.

Major points:

Why is it that both the expression of PrPc and infection with PrPsc lead to an increased sensitivity to

ferroptosis? As mentioned, the data supporting both claims is much stronger, but I feel that there is a critical mechanistic connection that has been overlooked. It would be of interest to determine how PrPSc is doing this but, admittedly, this is beyond the scope of the current work. However, I feel that it would be beneficial if the authors could speculate on this further, and incorporate PrPSc into the cartoon presented in Fig 6E.

Thanks for the comment. Certainly, this question has been an endeavor from the beginning of the project. We have considered several (non-mutually exclusive) explanations as to why PrPSc could induce ferroptosis, considering that PrPC is lost over time.

(1) PrPSc is perceived as an unfolded protein by cells, triggering GPX8 upregulation and mesenchymal sensitivity. How precisely this may happen is open to speculation, but retrograde transport would place PrPSc in the vicinity of GPX8.

(2) Individual cells sense PrPSc conversion and compensate by upregulating PrPC transiently, but then die by ferroptosis, resulting in global PrPC loss but protease-resistant PrPSc remains

(3) PrPSc readily induces membrane ROS, the final blow for cells that have upregulated the PrPC/GPX8/Rac3 axis. This lipid peroxidation would be consistent with published reports in mice.

As pointed out, it is difficult to consolidate these into this body of work, but it is important to set the stage for the next investigations. We therefore included our considerations in the Discussion and PrPSc into 6D.

Minor points:

Blots using CJD patient brains are inconsistently labeled. Are these each separate patients or from two patients as indicated in Supp Fig 8?

For human brain western blotting, every lane is a different brain sample. All blots show the same three control and same six sCJD brains. Sup Fig 8A was mislabeled with the sCJD subtypes that were loaded (CJD1 and CJD2 referred to subtype MM1 and MV2 respectively). We decided as there was no subtype differences observed (and subtype specific differences in pathogenesis was not the focus of this manuscript) that including this information on the blots would be confusing for readers unfamiliar with subtype specifics. Thus, these were all grouped together as sCJD samples for analysis. We have changed Sup Fig 8A to read sCJD matching the rest of the blots. Thank you for catching this mistake.

Line 85: "... indicating that PrPSc and RAC3 together provoke cellular loss". Do you mean accumulation of PrPSc and loss of RAC3?

Certainly, PrPSc acute toxicity is limited, and we did not specifically test PrPSc together with high expression of RAC3.

Line 355: "PrPC levels have been shown to decrease over the course of prion disease" lists reference 57. This is the incorrect reference for this claim. The correct reference: Mays CE, Kim C, Haldiman T, van der Merwe J, Lau A, Yang J, Grams J, Di Bari MA, Nonno R, Telling GC, Kong Q. Prion disease tempo determined by host-dependent substrate reduction. The Journal of clinical investigation. 2014 Jan 16;124(2).

I would suggest a careful checking of references.

Thanks for this point and the overall conclusions. We agree that the reviewer's reference is correct. We checked the others and made changes below.

52. Kocisko DA, et al. Cell-free formation of protease-resistant prion protein. *Nature* 370, 471-474 (1994).

This reference was removed.

55. Richt JA, et al. Production of cattle lacking prion protein. *Nature biotechnology* 25, 132-138 (2007).

And

56. Nuvolone M, et al. Strictly co-isogenic C57BL/6J-Prnp^{-/-} mice: A rigorous resource for prion science. *Journal of Experimental Medicine* 213, 313-327 (2016).

These references were replaced with:

Cell 1993 Jul 2;73(7):1339-47. doi: 10.1016/0092-8674(93)90360-3. Mice devoid of PrP are resistant to scrapie. H Büeler 1, A Aguzzi, A Sailer, R A Greiner, P Autenried, M Aguet, C Weissmann